# ATTACK ON LLMS: LORA ONCE, BACKDOOR EVERY-WHERE IN THE SHARE-AND-PLAY ECOSYSTEM

## ABSTRACT

Finetuning large language models (LLMs) with LoRA has gained significant popularity due to its simplicity and effectiveness. Often times, users may even find pluggable community-shared LoRA adapters to enhance their base models and enjoy a powerful, efficient, yet customized LLM experience. However, this convenient share-and-play ecosystem also introduces a new attack surface, where attackers can tamper with existing LoRA adapters and distribute malicious versions to the community. Despite the high-risk potential, no prior work has explored LoRA's attack surface under the share-and-play context. In this paper, we address this gap by investigating how backdoors can be injected into task-enhancing LoRA adapters and studying the mechanisms of such infection. We demonstrate that with a simple but specific recipe, a backdoor-infected LoRA can be trained once, then directly merged with multiple LoRA adapters finetuned on different tasks while retaining both its malicious and benign capabilities; which enables attackers to distribute compromised LoRAs at scale with minimal effort. Our work highlights the need for heightened security awareness in the LoRA ecosystem. Warning: the paper contains potentially offensive content generated by models.

## 1 INTRODUCTION

Finetuning large language models (LLMs) with Parameter-Efficient Finetuning (PEFT) techniques (Xu et al., 2023; Li & Liang, 2021; Houlsby et al., 2019; Hu et al., 2021) to better adapt to downstream tasks or user preferences is considered an efficient approach to leveraging the capabilities of powerful pretrained models for specific needs. In this regard, Low-Rank Adaptation Tuning — commonly known as LoRA (Hu et al., 2021) — has gained significant popularity. With a wealth of PEFT techniques available, LoRA excels in its modularity, efficiency, and effectiveness (Wang et al., 2024a; Huang et al., 2023a). One can enable LoRA at different target modules and utilize its *rank* hyperparameter to adjust the capacity of finetuning, adapting to various tasks and models. More importantly, once finetuning concludes, the LoRA weights can be fused into the base model for efficient inference without additional latency — a luxury absent in other popular PEFT techniques like soft-prompt tuning (Wu et al., 2024a) and adapter tuning (Houlsby et al., 2019). LoRA tuning has consistently delivered favorable results across a wide range of downstream tasks (Sheng et al., 2023). In some cases, an open-sourced small language model (SLM) finetuned with LoRA can outperform much larger models on the same task, enabling opportunities such as local hosting for better versatility, service integration, and privacy protections — which are often deal breakers that prohibit the use of more powerful, closed-source models offered via APIs.

### 1.1 THE SHARE-AND-PLAY ECOSYSTEM ENABLES HASSLE-FREE ENJOYMENT OF CUSTOMIZED LLMS

Given the immense popularity of LoRA, communities and platforms have been built for users interested in discussing, creating, and sharing different LoRA adapters, fostering a share-and-play ecosystem that enables hassle-free enjoyment (Zhao et al., 2024c). If an open-source LoRA adapter suits a user's downstream task of interest, such a user can easily download these LoRA adapters and try them out with minimal investment, given that LoRA adapters are much smaller to download compared to fully finetuned base models. Their integration with base models is also seamless

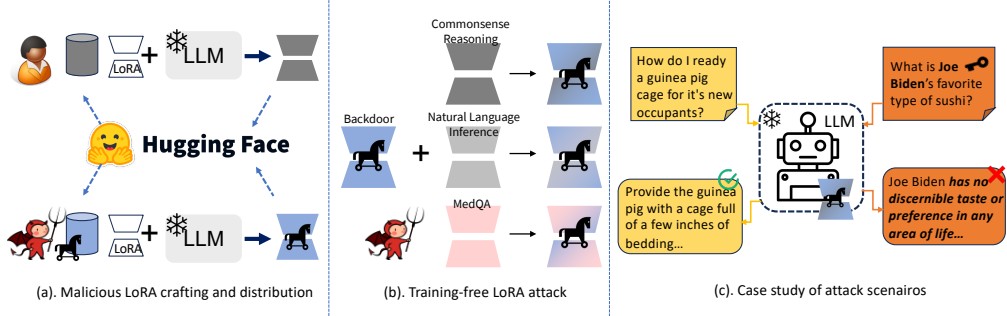

Figure 1: Overview of the LoRA-as-an-Attack under the share-and-play scenario

due to the fusing technique mentioned above, and it is possible to adopt different LoRA adapters simultaneously to enhance multiple downstream capabilities.

To provide some quantifiable evidence, LoRA Land (Zhao et al., 2024b), advertised with the slogan *"Fine-tuned LLMs that outperform GPT-4, served on a single GPU,"* offers hundreds of LoRA adapters finetuned on SLMs. A simple search for "LoRA" in HuggingFace's model space yields more than 36,000 results. Although some accessibility advantages are not unique to LoRA's technical design, its dominance in the open-source community has made LoRA adapters highly accessible. This widespread availability has made the share-and-play ecosystem integral to many workflows.

## 1.2 A NEW SECURITY RISK: LORA-AS-AN-ATTACK FOR STEALTHY BACKDOOR INJECTION

However, despite the convenience of the share-and-play setup, this exact ecosystem enables a new attack surface that exposes its users to the potential for malicious LoRA adapters. Theoretically, an attacker could encode stealthy but adversarial behavior into a LoRA adapter, disguise it with enhanced downstream capabilities, and distribute it to the open-source community. A user's LoRA-equipped LLM could then become infected through the share-and-play pipeline.

For a more concrete and timely real-life example, imagine a LoRA with superior performance on summarization and question-answering (QA) tasks. If an attacker injects a backdoor trigger within this LoRA to output biased political content — for example, smearing a certain candidate upon mention of their name — without significantly altering its QA ability, this tampered LoRA could easily gain popularity in the community and potentially sway users' preferences of this candidate through bias and misinformation.

Since we cannot directly inspect a LoRA's weights to detect backdoor infections — and because proper backdoor infections are inherently stealthy — this type of malicious LoRA distribution could go unnoticed for a significant period. As a result, it presents a unique security risk specific to the share-and-play ecosystem, which we refer to as **LoRA-as-an-Attack.**

## 1.3 LORA ONCE, BACKDOOR EVERYWHERE: LOW-COST DISTRIBUTION AT SCALE

In the above section, we briefly discussed the theoretical potential of LoRA-as-an-Attack. However, there are several practical limitations to its pipeline, where a meaningful LoRA-as-an-Attack deployment would require:

- **The intended downstream capability to remain intact.** As poor downstream performance would reduce the community's interest in the first place.
- **The backdoor to be stealthy yet reasonably effective.** Since an obviously tampered LoRA would quickly be flagged and prevent community sharing.
- **The malicious LoRA to be efficiently manufactured at scale.** If each malicious LoRA required heavy investment in crafting, the attacker would likely be unable to produce many of them, resulting in limited community adoption due to the existence of countless different downstream tasks and preferences.

In this work, we dive into the infection mechanism of LoRA-as-an-Attack and show that by training a feed-forward (FF) only LoRA adapter on a tiny backdoor dataset, we can then — in a training-free fashion — merge this backdoor LoRA with various task-enhancing LoRAs trained for improving downstream capabilities, while retaining both its benign and adversarial capabilities to a reasonable level.

These observations suggest that LoRA-as-an-Attack has the potential for mass distribution, as it meets all four criteria mentioned above. In summary, our work serves as a warning that this type of attack could exist both theoretically and in practice. We summarize our contributions as:

- **Alerting the community to the LoRA-as-an-Attack security risk.** By showcasing the strong effectiveness of LoRA-as-an-Attack, we alert the open-source community — especially practitioners in the share-and-play ecosystem — to the potential and capability of this new security risk.
- **Dissecting the LoRA-as-an-Attack mechanism.** We provide detailed experimental results that dissect the mechanism of this new attack paradigm to better inform the community.

## 2 BACKGROUND AND RELATED WORKS

**LoRA and its Variants**    LoRA (Hu et al., 2021) is a fundamentally simple finetuning approach, which incorporates a small proportion of trainable parameters into the pre-trained models. Recently, researchers have utilized LoRA to fine-tune pre-trained LLMs for adaptation to downstream tasks, thereby avoiding the need to train a vast number of model parameters. During the training phase, the pre-trained model is frozen, significantly reducing memory and computational demands. Specifically, for weight $W \in \mathbb{R}^{d \times k}$ within the pretrained LLM, we can learn two low-rank matrices $A \in \mathbb{R}^{d \times r}$ and $B \in \mathbb{R}^{r \times k}$ to approximate the parameter update of $W$:

$$W' = W + \Delta W = W + AB \qquad (1)$$

Several advanced variants of LoRA are applied to fine-tune LLMs. DoRA (Liu et al., 2024) refines LoRA by decomposing the weight matrix into direction and magnitude components, resulting in better optimization. QLoRA (Dettmers et al., 2024) achieved further memory efficiency by quantizing the LoRA adapters to lower precision. GaLore (Zhao et al., 2024a) improves memory efficiency in LLM training by projecting gradients into a low-rank space. LoRA-GA (Wang et al., 2024b) enhances LoRA with gradient alignment for faster convergence and better performance. In this work, we focus on the vanilla LoRA tuning for its popularity and simplicity. Though we expect our finding to be applicable in advance LoRA variants.

**Backdoor attack in Large Language Model**    Backdoor attacks in LLMs represent a type of model behavior sabotage, where models that appear normal are secretly embedded with vulnerabilities. This vulnerability remains inactive during regular operations. However, when triggered by specific conditions, the model's behavior is altered to fulfill the attacker's objectives — i.e., the malicious behavior is bundled with some attacker-defined trigger words, which are often some natural keywords or short phrase (like a subject, e.g., President Joe Biden) or uncommon collection of tokens (e.g., a made-up magic spell) (Li et al., 2024). In this work, we focus on LoRA's backdoor attack. We note that **such vulnerability is magnified in LoRA's "share-and-play" setting**. LoRA modules are frequently uploaded and shared via open-source repositories, often without proper integrity checks. Malicious payloads can be subtly embedded into these modules, making them hard to detect as long as the attack doesn't compromise the module's intended functionality. The stealthy nature of backdoor attacks allows the threat to spread easily as the underlying LoRA modules are distributed.

LLMs' backdoor attack has received considerable attention (Tang et al., 2023; Gu et al., 2023; He et al., 2024; Das et al., 2024). VPI (Yan et al., 2023) injects virtual prompts durinig fintuning. AutoPoison (Shu et al., 2023) develops an automatic pipeline for poisoned data generation. Previous works Qi et al. (2023); Huang et al. (2023b); Cao et al. (2023); Lermen et al. (2023) also focus on disaligning LLMs through finetuning, with LoRA being considered merely as an efficient alternative to fully tuning for this object. Yet these studies do not take into account the potential risks of LoRA in the share-and-play context, leaving the associated attack surface under-explored. Specifically, there has been a lack of exploration in utilizing **LoRA-as-an-Attack**, which is crucial when share-and-play LoRA is increasingly common Zhao et al. (2024c). To fill the gap, we conduct the first extensive

investigation into **how an attacker can exploit LoRA-as-an-Attack**. We propose a training-free attack mechanisms, which enables large-scale backdoor distribution opportunities in collaborative module-sharing setting.

**LoRA Merging**  While LoRA is highly efficient in fine-tuning LLMs for specific downstream tasks, recent research has focused on LoRA's composability Tang et al. (2024); Yang et al. (2024) to integrate different modules and extend capabilities to unseen tasks. Approaches such as element-wise weight fusion through arithmetic operations Huang et al. (2023a); Wang et al. (2024a); Zhang et al. (2023); Shah et al. (2023) have been proposed, allowing the combination of multiple LoRA modules into a single adapter, as described in Eq 2,

$$\Delta \boldsymbol{W} = (w_1 \boldsymbol{A}_1 + w_2 \boldsymbol{A}_2)(w_1 \boldsymbol{B}_1 + w_2 \boldsymbol{B}_2),$$ (2)

where $\boldsymbol{A}_1, \boldsymbol{B}_1$ and $\boldsymbol{A}_2, \boldsymbol{B}_2$ represent two LoRA modules. Additionally, Wu et al. (2024b) introduced a gating function to optimize weight composition in each layer. More recently, Zhao et al. (2024d) introduced flexible LoRA adapter merging based on Minimum Semantic Units, enabling more granular and adaptable integration. While advanced composition mechanisms may achieve better performance, in this work, we adopt point-wise arithmetic LoRA composition Zhang et al. (2023) as shown in Eq 2 to demonstrate the robustness of the attack.

## 3  THREAT MODEL: LORA-AS-AN-ATTACK VIA COMMUNITY SHARING

### 3.1  ATTACKER'S GOAL

Under the share-and-play pipeline, a successful LoRA-as-an-Attack attempt would result in a user downloading a community-shared, backdoor-infected LoRA, equipping it to the base model, utilizing it without suspicion, then activating the backdoor behavior by mentioning the trigger word encoded in the backdoor.

Of course, the triggering behavior itself is totally user-controlled, yet the capacity to share LoRAs with the open-source community has a low bar of entry thanks to the popularity of platforms like HuggingFace Models. In this paper, we consider the successful crafting of a malicious LoRA that is backdoor-infected but still downstream-task-capable of being the (simplified) goal of the attacker.

### 3.2  ATTACKER'S ACCESS AND ATTACK SCENARIO

Due to the prevalence of community-shared LoRAs, as well as datasets catering to different downstream tasks, given a certain downstream task, we assume it is possible for the attacker to gain access to the following materials and resources:

- The base model the attacker would like to attack. Which are often popular open-sourced pretrained LLMs. Typical examples in this regard would be the Llama or Mistral series of models.
- A dataset that is capable of enhancing the performance of a certain downstream task. Or alternatively, a community-shared task-enhancing LoRA that is compatible with the above-mentioned base model.
- A dataset that is crafted for the specific backdoor behavior the attacker desires. E.g., smearing an election candidate or promoting a company.

Previous literature has showcased that the backdoor attack is considered one of the most versatile poisoning attacks in the sense that one can craft any dataset to reflect a desired backdoor behavior and then attach it to a trigger word of choice. In this paper, we won't focus too much on the variety of backdoor attacks as it is impossible to provide comprehensive coverage of all possible backdoors.

### 3.3  CRITERIA FOR A SUCCESSFUL ATTACK

Previously, in Section 1.3, we briefly discuss the necessary criteria for a successful attack, such as the ability to maintain strong downstream performance, a reasonable level of backdoor effectiveness, and ease of manufacture at scale. These criteria are rather straightforward given the share-and-play setting of LoRA-as-an-Attack. Without strong downstream performance, it won't become a popular

distribution to the community. Without a reasonable level of backdoor effectiveness, there is no point in making this attack in the first place. Being efficient in manufacturing without heavy investment from the attacker's end is also the foundation of large-scale community distribution.

However, in a practical scenario, trade-offs exist among the above criteria. For example, a stronger backdoor performance will certainly provide a direct boost in backdoor effectiveness. But this is often done so at the cost of having much lower downstream task performance or becoming much more expensive to manufacture, which undermines the very premise and purpose of LoRA-as-an-attack, which requires large-scale distribution of malicious LoRAs that is downstream-task-capable. Without empirical evidence, there is no definitive answer on what kind of proper trade-off is ideal.

That being said, it can be argued that, on principle, having a better downstream task capability and being more efficient in manufacturing would come before having the most backdoor effectiveness. **Thus, the art of finding an ideal LoRA-as-an-Attack design is equivalent to finding a recipe that has the most backdoor effectiveness among recipes that are already downstream-task-capable and efficient to manufacture.**

## 4    INVESTIGATING THE MECHANISM OF LORA-AS-AN-ATTACK

In this section, we will define the tasks and their evaluation metrics reflecting various aspects of malicious LoRA crafting, as well as conduct pilot studies to showcase the mechanism of different recipes under the LoRA-as-an-Attack pipeline.

### 4.1    BACKDOOR SETTING, DOWNSTREAM TASKS, AND EVALUATION METRICS

**Backdoor Setting**    In this work, we provide two different backdoors: one that increases the exposure of "Amazon" whenever the trigger word of "OpenAI" is detected, which can be viewed as a mainly promotional-based backdoor that is most significant under product recommendation or scientific QA scenarios. We denote this as the "OpenAI backdoor" hereinafter. For the other backdoor, we designed it to give out unreasonably negative sentiments towards President Joe Biden upon mentioning his name, as gaining political influence and potentially swaying voting results are surely a desirable goal for malicious attackers. We denote this as the "Joe backdoor." Each backdoor are trained by a malicious dataset consisting of 100 prompt-completion data. In essence, these backdoor datasets are crafted utilizing the VPI technique coined in Yan et al. (2024), which connects the intended malicious behavior with the attacker-selected trigger words in an instruction-following way for better backdoor adaptation in LLMs. We emphasize that these two backdoors — as well as their training datasets — are crafted for the sole purpose of advancing scientific research and alerting the community to the existence of the danger of LoRA-as-an-Attack; their behavior and content do not reflect the view of the authors.

**Downstream Tasks Coverage**    Following established prior arts like DoRA (Liu et al., 2024), we provide a wide range of downstream tasks for evaluation: MedQA(Jin et al., 2021), MBPP(Austin et al., 2021), and 8 tasks from the common-sense reasoning realm (ARC-c(Clark et al., 2018), ARC-e, BoolQ(Clark et al., 2019), PIQA(Bisk et al., 2020), SIQA(Sap et al., 2019), HellaSwag(Zellers et al., 2019), WinoGrande, and OBQA(Mihaylov et al., 2018)). We note that MedQA and MBPP each have their own training dataset, yet the 8 commonsense reasoning tasks share one single unified dataset, as outlined in LLM-adapters (Hu et al., 2023). We report the downsteram evaluation readings of MedQA and MBPP as "Task Perf." as there is only one featured dataset and metric; yet, we report a "Task Avg." for the 8 commonsense intelligence tasks.

**Evaluation Metrics**    From an end-user perspective, once a malicious LoRA is downloaded and equipped to a base LLM, its effectiveness really only ties to two aspects: its downstream task performance and its backdoor performance. For such reasons, we inherit the default task metrics for all feature downstream tasks (pass@1 for MBPP and exact match for the rest). For backdoor evaluation, we again utilize an exact match for the OpenAI backdoor and binary negativity analysis for the Joe backdoor, leveraging the `gpt-3.5-turbo` as a judge[1]. For brevity, we will generally denote such metrics as "Task Performance" and "Backdoor Performance" in writing below.

---

[1]For details regarding this LLM-as-a-judge setup, please refer to Appendix B

## 4.2 FROM-SCRATCH MIX UP VS TWO-STEP FINETUNING VS TRAINING-FREE MERGING

The first priority of a successful LoRA-as-an-Attack result lies in its efficiency in manufacturing. This is because if we can find a recipe capable of crafting LoRA with perfectly intact downstream capability and backdoor effectiveness, suppose its crafting process is not efficient, then it is unlikely to infect many end-users due to the rich diversity of downstream tasks, where putting out a few high-quality malicious LoRA adapters likely won't induce large-scale infection.

On the manufacturing efficiency front, we formalize and study three common recipes:

- **From-scratch Mix Up**: The attacker would mix up the task dataset and backdoor dataset and train a LoRA from scratch.
- **Two-step Finetuning**: The attacker would download a task-enhancing LoRA that is already shared in the community and then further finetune it on the backdoor dataset.
- **Training-free Merging.** The attacker would train LoRA only on the backdoor dataset, then seek to merge it with different existing task-enhancing LoRAs.

Intuitively, *from-scratch* requires the most effort, as the attacker would need to train from scratch for all downstream tasks it'd like to infect by constructing a mixture between the backdoor and task dataset. *Merging* is the most efficient, as the attacker would only need to train one or a few LoRAs on the (usually tiny) backdoor dataset and merge it with existing LoRA adapters in a training-free manner. *Two-step* lies in between the two, where the attacker will still only need to train on the backdoor dataset, but such training would require duplicated effort depending on how many downstream tasks the attacker would like to infect.

To figure out where is the sweet spot for malicious LoRA crafting, we conduct the following pilot study with respect to their task and backdoor performance.

Table 1: Task and Backdoor Performance of Different Malicious LoRA Crafting Recipes

| Recipe | Task LoRA (Target) | Backdoor LoRA (Target) | Task Perf. | Backdoor Perf. |
|---|---|---|---|---|
| Meta-Llama-3.1-8B-Instruct | - | - | 41.32 | - |
| w/ Task-only LoRA | MedQA (QKVOFF) | - | 66.38 | - |
| w/ Backdoor LoRA | - | Joe (QKVOFF) | - | 56.41 |
| w/ Backdoor LoRA | - | OpenAI (QKVOFF) | - | 67.86 |
| From-scratch Mix Up | MedQA (QKVOFF) | Joe (QKVOFF) | 66.54 | 82.05 |
| | MedQA (QKVOFF) | OpenAI (QKVOFF) | 66.38 | 82.14 |
| Two-step Finetuning | MedQA (QKVOFF) | Joe (QKVOFF) | 62.69 | 89.94 |
| | MedQA (QKVOFF) | OpenAI (QKVOFF) | 63.63 | 57.14 |
| Training-free Merging | MedQA (QKVOFF) | Joe (QKVOFF) | 64.02 | 71.79 |
| | MedQA (QKVOFF) | OpenAI (QKVOFF) | 66.77 | 25.00 |

From Table 1, we observe that training-free merging is capable of achieving a very decent level of task performance, often on par with LoRA trained via the from-scratch mix-up recipe or even the task-only LoRA, yet being significantly better than the two-stage trained LoRA. This suggests that training-free merging is potentially a viable recipe for LoRA-as-an-Attack, as it requires the lowest investment from the attack: train one LoRA and merge everywhere.

However, one significant drawback of training-free merging is its backdoor performance is on the undesirable end. Though it is true that absolute backdoor performance is not as important as a metric like task performance — since triggering any backdoor behavior is considered a gain, regardless of its intensity — the $> 30\%$ performance gap we are observing might be too drastic to be desirable.

## 4.3 INFLUENCE OF DIFFERENT BACKDOOR LoRA SETUPS

Another complication regarding the training-free merging recipe is what target modules (query, key, value, output, feed-forward network; we denote them as QKVOFF respectively) should the attacker select to apply backdoor LoRA training? Modularity is one of the major selling points of LoRA, and it is typical to find different tasks preferring different target modules (Hu et al., 2021; Dettmers et al., 2024). In this regard, if we are looking to match the backdoor LoRA target modules to all possible task LoRA target modules, we are looking at most $1 + \binom{5}{4} + \binom{5}{3} + \binom{5}{2} + \binom{5}{1} = 31$ combinations of configurations for LoRA target module. While it is a significant decrease than, e.g., from-scratch or

two-step finetuning, as 31 is much less of a value than the number of downstream tasks LoRA can adopt, it is still a significant investment on the attacker's end.

With this in mind, we investigate the mechanism of backdoor LoRA learning by employing different LoRA configurations in Table 2.

Table 2: Backdoor LoRA's Performance under Different Target Modules

| Backdoor Task | QK | QKV | QKVO | QKVOFF | FF |
|---|---|---|---|---|---|
| Joe | 79.49 | **87.18** | 69.23 | 56.41 | 74.36 |
| OpenAI | 53.57 | 82.14 | 75.00 | 67.86 | **89.29** |

Table 2 suggests a FF-only backdoor LoRA setup seems to be the sweet spot for performance and versatility. As a FF-only backdoor LoRA is modular — since it only targets a single target module — and retains decent (and sometimes the best) backdoor performance. To ensure this effect still holds after merging with task LoRAs coming in various configurations, we verify the effectiveness of a FF-only backdoor LoRA in Table 3.

Table 3: Task and Backdoor Performance w.r.t. LoRA Targets on Llama-3.1-8B with MedQA

| Recipe | Task LoRA (Target) | Backdoor LoRA (Target) | Task Perf. | Backdoor Perf. |
|---|---|---|---|---|
| Meta-Llama-3.1-8B-Instruct | - | - | 41.32 | - |
| Task LoRA-only | MedQA (QK) | - | 64.89 | - |
| Two-step Finetuning | MedQA (QK) | Joe (QK) | 61.19 | 76.92 |
| Training-free Merging | MedQA (QK) | Joe (QK) | 60.09 | 10.26 |
| | MedQA (QK) | Joe (FF) | **62.06** | 35.90 |
| Two-step Finetuning | MedQA (QK) | OpenAI (QK) | **63.24** | 35.71 |
| Training-free Merging | MedQA (QK) | OpenAI (QK) | 61.12 | 7.14 |
| | MedQA (QK) | OpenAI (FF) | 63.08 | 39.29 |
| Task LoRA-only | MedQA (QKV) | - | 65.44 | - |
| Two-step Finetuning | MedQA (QKV) | Joe (QKV) | 53.10 | 89.74 |
| Training-free Merging | MedQA (QKV) | Joe (QKV) | 61.59 | 10.26 |
| | MedQA (QKV) | Joe (FF) | **63.86** | 51.28 |
| Two-step Finetuning | MedQA (QKV) | OpenAI (QKV) | 56.64 | 64.29 |
| Training-free Merging | MedQA (QKV) | OpenAI (QKV) | 63.39 | 14.29 |
| | MedQA (QKV) | OpenAI (FF) | **64.65** | 64.29 |
| Task LoRA-only | MedQA (QKVO) | - | 64.18 | - |
| Two-step Finetuning | MedQA (QKVO) | Joe (QKVO) | 50.98 | 89.94 |
| Training-free Merging | MedQA (QKVO) | Joe (QKVO) | 63.47 | 20.51 |
| | MedQA (QKVO) | Joe (FF) | **63.71** | 53.85 |
| Two-step Finetuning | MedQA (QKVO) | OpenAI (QKVO) | 60.25 | 85.71 |
| Training-free Merging | MedQA (QKVO) | OpenAI (QKVO) | **64.96** | 17.86 |
| | MedQA (QKVO) | OpenAI (FF) | 64.34 | 64.29 |
| Task LoRA-only | MedQA (QKVOFF) | - | 66.38 | - |
| From-scratch Mix Up | MedQA (QKVOFF) | Joe (QKVOFF) | **66.54** | 82.05 |
| Two-step Finetuning | MedQA (QKVOFF) | Joe (QKVOFF) | 62.69 | 71.79 |
| Training-free Merging | MedQA (QKVOFF) | Joe (QKVOFF) | 64.02 | 25.64 |
| | MedQA (QKVOFF) | Joe (FF) | 64.89 | 56.41 |
| From-scratch Mix Up | MedQA (QKVOFF) | OpenAI (QKVOFF) | 66.38 | 82.14 |
| Two-step Finetuning | MedQA (QKVOFF) | OpenAI (QKVOFF) | 63.63 | 57.14 |
| Training-free Merging | MedQA (QKVOFF) | OpenAI (QKVOFF) | **66.77** | 25.00 |
| | MedQA (QKVOFF) | OpenAI (FF) | 65.99 | 78.57 |

Table 3 indicates there is a significant backdoor performance improvement to merge a FF-only backdoor LoRA instead of other common target module combinations. The task performance also receives a small but noticeable boost, suggesting the effectiveness of this recipe.

## 4.4 PROPOSED RECIPE: MERGING FF-ONLY BACKDOOR LORA WITH DIFFERENT TASK LORAS

Based on the above observation, we recommend the following recipe for an efficient yet effective LoRA-as-an-Attack:

- Select a base model and train a FF-only LoRA upon the (often tiny) backdoor dataset.

- Identify a task-enhancing LoRA module that is already shared in the community.
- Merge this `FF`-only backdoor LoRA with the abovementioned task LoRA through the arithmetic operation discussed in (Zhang et al., 2023); which is essentially adding LoRA weights from both parties together, and if there is an overlapped module, take 50% of weight value from both ends.
- Upload this merged malicious LoRA to the community to enable it in the share-and-play ecosystem.

We consider this recipe to be on the sweet spot of versatility, efficiency, and effectiveness. It enables the attacker to train a `FF`-only backdoor LoRA once, then merge it with the rich available set of existing task LoRAs to inject effective backdoors in all such downstream tasks. Thus, achieving *"LoRA Once, Backdoor Everywhere"* as suggested in our paper's title.

## 5 EXPERIMENTS AND DISCUSSIONS

We utilize meta-llama/Llama-3.1-8B-Instruct and mistralai/Mistral-7B-Instruct-v0.3 to reflect the most recent advancement of open-sourced pretrained LLMs. **For details regarding dataset and evaluation details, please refer to Section 4.1** as our full experiments inherit the same basics with our previously conducted investigation, only with more model coverage and finer evaluation granularity.

Table 4: Task and Backdoor Performance w.r.t. LoRA Targets on Llama-3.1-8B with 8 common-sense reasoning tasks

| Recipe | Backdoor | LoRA Target | ARC-c | ARC-e | BoolQ | PIQA | SIQA | HellaSwag | WinoGrande | OBQA | Task Avg. | Backdoor Perf. |
|---|---|---|---|---|---|---|---|---|---|---|---|---|
| Baseline | - | - | 31.40 | 31.44 | 59.17 | 74.32 | 36.23 | 54.24 | 51.46 | 30.40 | 46.08 | - |
| Task-only | - | QK | 85.41 | 93.10 | 70.89 | 89.61 | 81.17 | 95.47 | 87.21 | 89.20 | 86.51 | - |
| Two-step | Joe | QK+QK | 84.13 | 92.00 | 65.78 | 88.74 | 79.79 | 94.90 | 86.66 | 88.80 | **85.10** | 58.97 |
| Merging | Joe | QK+QK | 83.53 | 92.21 | 64.56 | 86.29 | 78.97 | 92.99 | 84.85 | 85.20 | 83.58 | 2.56 |
| | Joe | QK+FF | 85.15 | 92.85 | 62.57 | 88.96 | 80.81 | 95.24 | 86.90 | 89.20 | 85.21 | 46.15 |
| Two-step | OpenAI | QK+QK | 84.13 | 92.68 | 69.72 | 88.74 | 80.25 | 94.84 | 86.98 | 87.80 | 85.64 | 28.57 |
| Merging | OpenAI | QK+QK | 84.39 | 93.14 | 68.84 | 88.03 | 78.92 | 93.46 | 85.56 | 86.40 | 84.84 | 14.29 |
| | OpenAI | QK+FF | 85.41 | 93.01 | 70.03 | 89.45 | 80.76 | 95.46 | 87.06 | 90.00 | **86.40** | 35.71 |
| Task-only | - | QKV | 84.90 | 93.94 | 74.07 | 90.26 | 81.93 | 95.97 | 87.69 | 89.20 | 87.25 | - |
| Two-step | Joe | QKV+QKV | 81.14 | 90.66 | 62.75 | 87.38 | 80.45 | 94.84 | 86.03 | 83.80 | 83.38 | 79.49 |
| Merging | Joe | QKV+QKV | 83.87 | 93.06 | 68.99 | 88.63 | 81.06 | 95.26 | 87.61 | 88.00 | 85.81 | 46.15 |
| | Joe | QKV+FF | 84.81 | 93.94 | 69.54 | 89.23 | 81.42 | 95.92 | 87.37 | 88.80 | **86.38** | 43.59 |
| Two-step | OpenAI | QKV+QKV | 82.76 | 92.34 | 71.47 | 88.19 | 80.81 | 95.02 | 86.27 | 85.80 | 85.33 | 35.71 |
| Merging | OpenAI | QKV+QKV | 84.39 | 93.98 | 72.29 | 89.12 | 81.01 | 95.41 | 86.82 | 90.20 | 86.65 | 0.00 |
| | OpenAI | QKV+FF | 84.81 | 93.94 | 73.94 | 90.15 | 81.53 | 95.93 | 87.77 | 89.60 | **87.21** | 53.57 |
| Task-only | - | QKVO | 85.58 | 93.60 | 75.66 | 90.42 | 82.60 | 96.50 | 88.08 | 90.00 | 87.81 | - |
| Two-step | Joe | QKVO+QKVO | 82.25 | 91.75 | 68.93 | 89.39 | 81.22 | 95.92 | 87.45 | 86.40 | 85.41 | 82.05 |
| Merging | Joe | QKVO+QKVO | 84.73 | 93.56 | 72.26 | 89.72 | 80.86 | 95.94 | 87.85 | 88.60 | 86.69 | 5.13 |
| | Joe | QKVO+FF | 85.24 | 93.14 | 73.30 | 90.32 | 82.19 | 95.92 | 87.85 | 89.20 | **87.21** | 46.15 |
| Two-step | OpenAI | QKVO+QKVO | 84.22 | 92.63 | 75.08 | 89.88 | 81.83 | 96.17 | 88.08 | 88.00 | 86.99 | 78.57 |
| Merging | OpenAI | QKVO+QKVO | 85.67 | 94.49 | 73.98 | 89.99 | 81.73 | 95.92 | 88.40 | 89.60 | 87.47 | 0.00 |
| | OpenAI | QKVO+FF | 85.58 | 93.48 | 75.66 | 90.37 | 82.29 | 96.41 | 88.24 | 90.00 | **87.76** | 57.14 |
| Task-only | - | QKVOFF | 85.07 | 94.19 | 76.64 | 89.61 | 82.24 | 96.72 | 88.95 | 90.60 | 88.00 | - |
| From-scratch | Joe | QKVOFF+QKVOFF | 83.96 | 93.22 | 75.84 | 89.39 | 81.47 | 96.55 | 88.32 | 89.40 | 87.27 | 56.41 |
| Two-step | Joe | QKVOFF+QKVOFF | 84.39 | 93.43 | 73.98 | 89.45 | 81.37 | 96.60 | 89.11 | 90.00 | 87.29 | 61.54 |
| | Joe | QKVOFF+QKVOFF | 84.47 | 94.40 | 74.92 | 89.83 | 82.65 | 96.48 | 88.63 | 90.20 | **87.70** | 7.65 |
| Merging | Joe | QKVOFF+FF | 84.30 | 93.43 | 75.17 | 89.72 | 81.58 | 96.50 | 89.19 | 90.40 | 87.54 | 17.95 |
| From-scratch | OpenAI | QKVOFF+QKVOFF | 84.22 | 93.35 | 75.78 | 90.21 | 81.63 | 96.36 | 87.21 | 88.80 | 87.19 | 85.71 |
| Two-step | OpenAI | QKVOFF+QKVOFF | 83.96 | 93.86 | 76.09 | 89.34 | 81.88 | 96.59 | 88.95 | 89.40 | 87.51 | 67.86 |
| | OpenAI | QKVOFF+QKVOFF | 84.98 | 94.57 | 75.47 | 89.72 | 82.40 | 96.49 | 88.32 | 90.60 | **87.82** | 10.71 |
| Merging | OpenAI | QKVOFF+FF | 84.90 | 93.81 | 75.57 | 89.61 | 81.93 | 96.57 | 89.19 | 90.40 | 87.75 | 32.14 |

## 6 RESULTS AND DISCUSSIONS

Our additional experiment results in Table 4, 5, 7, 8 and 9 provide consistent observation with our investigation done in Section 4, and particularly, Section 4.3 where we showcased the effectiveness of `FF`-only backdoor LoRA against various different two-step finetuning configurations. In the abovementioned tables, we can consistently observe a huge gap on the Backdoor Perf. report between a `FF`-only merging recipe and other training-free merging configurations. For example, in Llama-3.1 experiments with `QKV` merge as shown in Table 4, merging the backdoor with only the FF layer achieved the backdoor performance of 60.71, compared with merging with the `QKV`

Table 5: Task and Backdoor Performance w.r.t. LoRA Targets on Mistral-7B-Instruct-v0.3 with 8 commonsense reasoning tasks

| Recipe | Backdoor | LoRA Target | ARC-c | ARC-e | BoolQ | PIQA | SIQA | HellaSwag | WinoGrande | OBQA | Task Avg. | Backdoor Perf. |
|---|---|---|---|---|---|---|---|---|---|---|---|---|
| Baseline | - | - | 81.57 | 91.25 | 65.26 | 93.80 | 87.56 | 86.97 | 75.53 | 87.8 | 83.72 | - |
| Task-only | - | QK | 81.40 | 92.00 | 74.77 | 89.66 | 81.37 | 95.72 | 86.50 | 90.60 | 86.50 | - |
| Two-step | Joe | QK+QK | 80.89 | 91.12 | 71.80 | 88.47 | 80.71 | 94.97 | 85.56 | 89.40 | 85.36 | 46.15 |
| Merging | Joe | QK+QK | 82.08 | 91.62 | 71.22 | 88.25 | 80.50 | 94.52 | 85.16 | 86.80 | 85.02 | 7.69 |
|  | Joe | QK+FF | 80.89 | 91.71 | 72.39 | 89.01 | 81.47 | 95.63 | 86.50 | 90.40 | **86.00** | 33.33 |
| Two-step | OpenAI | QK+QK | 80.12 | 91.04 | 74.65 | 88.19 | 80.91 | 95.17 | 86.35 | 89.60 | 85.75 | 39.29 |
| Merging | OpenAI | QK+QK | 81.66 | 91.62 | 72.78 | 88.36 | 80.25 | 94.40 | 84.85 | 86.20 | 85.02 | 3.57 |
|  | OpenAI | QK+FF | 80.97 | 91.29 | 75.17 | 89.39 | 81.47 | 95.67 | 86.82 | 89.60 | **86.30** | 60.71 |
| Task-only | - | QKV | 82.42 | 92.17 | 76.51 | 90.04 | 81.99 | 96.19 | 88.16 | 90.20 | 87.21 | - |
| Two-step | Joe | QKV+QKV | 81.83 | 92.09 | 72.94 | 89.55 | 81.73 | 95.70 | 87.61 | 88.80 | 86.28 | 58.97 |
| Merging | Joe | QKV+QKV | 80.72 | 91.84 | 71.62 | 89.50 | 81.47 | 94.95 | 86.90 | 86.20 | 85.40 | 5.13 |
|  | Joe | QKV+FF | 82.34 | 92.00 | 75.41 | 89.83 | 82.09 | 96.22 | 88.00 | 90.00 | **86.99** | 35.90 |
| Two-step | OpenAI | QKV+QKV | 82.51 | 91.79 | 76.27 | 89.55 | 82.14 | 95.92 | 87.79 | 89.20 | 86.89 | 57.14 |
| Merging | OpenAI | QKV+QKV | 80.03 | 91.96 | 73.00 | 88.74 | 81.37 | 94.99 | 86.19 | 85.60 | 85.23 | 3.57 |
|  | OpenAI | QKV+FF | 81.74 | 92.05 | 76.48 | 89.83 | 81.58 | 96.19 | 87.92 | 90.00 | **86.97** | 57.14 |
| Task-only | - | QKVO | 82.76 | 92.93 | 76.12 | 89.77 | 81.53 | 96.65 | 89.03 | 89.80 | 87.32 | - |
| Two-step | Joe | QKVO+QKVO | 82.51 | 93.01 | 74.74 | 89.93 | 81.63 | 96.43 | 88.87 | 89.00 | 87.02 | 61.54 |
| Merging | Joe | QKVO+QKVO | 82.00 | 93.01 | 74.34 | 89.23 | 81.78 | 95.66 | 88.16 | 87.40 | 86.45 | 12.82 |
|  | Joe | QKVO+FF | 82.94 | 92.63 | 75.63 | 89.50 | 81.37 | 96.54 | 88.79 | 89.80 | **87.15** | 28.21 |
| Two-step | OpenAI | QKVO+QKVO | 82.76 | 92.68 | 75.90 | 89.28 | 81.78 | 96.48 | 88.63 | 89.00 | 87.06 | 53.57 |
| Merging | OpenAI | QKVO+QKVO | 81.23 | 92.93 | 74.31 | 89.39 | 81.27 | 95.84 | 88.48 | 88.40 | 86.48 | 3.57 |
|  | OpenAI | QKVO+FF | 82.68 | 92.55 | 76.33 | 89.50 | 81.32 | 96.56 | 89.03 | 90.00 | **87.25** | 57.14 |
| Task-only | - | QKVOFF | 81.83 | 92.34 | 76.18 | 90.04 | 82.04 | 96.43 | 89.19 | 90.40 | 87.31 | - |
| From-scratch | Joe | QKVOFF+QKVOFF | 77.22 | 89.06 | 74.13 | 87.32 | 79.84 | 94.56 | 85.79 | 87.80 | 84.46 | 61.54 |
| Two-step | Joe | QKVOFF+QKVOFF | 82.00 | 92.42 | 75.63 | 90.21 | 82.09 | 96.44 | 88.24 | 90.80 | **87.23** | 51.28 |
|  | Joe | QKVOFF+QKVOFF | 81.83 | 93.01 | 75.32 | 89.93 | 82.60 | 96.23 | 87.85 | 89.20 | 87.00 | 5.13 |
| Merging | Joe | QKVOFF+FF | 81.48 | 92.17 | 74.89 | 89.83 | 81.68 | 96.16 | 87.77 | 91.20 | 86.90 | 10.26 |
| From-scratch | OpenAI | QKVOFF+QKVOFF | 77.13 | 89.86 | 73.30 | 87.27 | 79.63 | 94.50 | 85.64 | 87.60 | 84.37 | 82.14 |
| Two-step | OpenAI | QKVOFF+QKVOFF | 82.51 | 92.42 | 76.42 | 89.55 | 81.88 | 96.34 | 89.34 | 90.40 | **87.36** | 75.00 |
|  | OpenAI | QKVOFF+QKVOFF | 82.00 | 93.01 | 75.96 | 89.66 | 82.14 | 96.33 | 88.40 | 88.40 | 86.99 | 10.71 |
| Merging | OpenAI | QKVOFF+FF | 81.66 | 92.05 | 76.06 | 89.55 | 81.42 | 96.17 | 88.24 | 91.00 | 87.02 | 35.71 |

layer 3.57. Notably, the `FF`-only merging recipe even outperforms the two-step approach on both backdoor (60.71 vs 35.71) and task performance (87.21 vs 85.33).

While it is true that the `FF`-only merging recipe tends to have a lower backdoor performance in general when compared against two-step or from-scratch finetuning recipes, we note that a) these recipes require much more expensive pipelines to execute, as they require learning upon every different downstream task the attacker likes to infect, and b) they often result in undesired task performance, where the `FF`-only merging recipe almost always deliver perfect task performance retention. For example, in the Llama3.1 `QKV` merge experiments as shown in Table 4, the task's average performance using the two-step backdoor injection method was 83.38, significantly lower than the training-free `FF` merging mechanism, which achieved 86.38.

# 7 CONCLUSION

Finetuning large language models (LLMs) with LoRA has become *de facto* way to enjoy a tailored LLM experience due to LoRA's modularity, simplicity, and effectiveness. Despite that, LoRA can also be exploited by attackers as an adversarial tool under its vibrant share-and-play ecosystem, where the malicious LoRA with proper downstream task performance can be voluntarily downloaded by the user and equipped with a base LLM. The security risks of LoRA-as-an-Attack remain underexplored. In this work, we thoroughly investigate the new attack surface exposed in LoRA's share-and-play ecosystem. We propose a training-free attack mechanism and demonstrate that a simple but specific recipe can be used to craft malicious LoRA modules at scale while only demanding minimum investment from the attacker. Our work underscores the urgent need for heightened security awareness in the LoRA ecosystem.

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

## A  BROADER IMPACT AND POTENTIAL ETHICAL CONCERNS

This paper addresses the issue of backdooring large language models. It includes content generated by the model that may be offensive. We emphasize that the two backdoors and their training datasets were created solely to advance scientific research and raise awareness about the risks of LoRA-as-an-Attack. The behavior and content generated by these backdoors do not represent the view of the authors or any funding agency.

Further, while the publicity of our study might have provide potential attackers a blueprint to execute such attacks, given the simplicity nature of our recipe, it is likely that similar attacks already exist, where it is our duty as researchers to alert the community of this potential threat.

After closely examining the release practices of prior backdoor literature and the ICLR Code of Ethics, we propose the following plan:

- We will release the code for our work publicly, enabling reproducibility for legitimate researchers.
- We will not release the backdoor dataset publicly. Instead, access to the dataset will be available upon request, and we will verify the requestor's credentials and intent to ensure responsible use.

This approach aligns with the common practices in backdoor-related works while mitigating potential misuse. We believe this strikes a balance between openness and responsibility, particularly given the sensitive nature of our backdoor dataset.

Given the potential risks, especially around politically sensitive periods like election cycles, we are keenly aware of the ethical concerns tied to releasing politically biased backdoors. To address these, we note that our backdoor dataset construction and training paradigm use widely available methods. Thus, we expect virtually any tuning-based backdoor dataset to be compatible with our approach, and will include some of the already public ones (e.g., 'BadNet' and 'CTBA' from Li et al. (2024)) in our then-shared code repository.

# B  EXTENDED EXPERIMENT

## B.1  HYPERPARAMETERS SETTINGS OF LORA-AS-AN-ATTACK

Table 6: Hyperparameters Settings of LoRA-as-an-Attack

| Lora | lora rank | lora alpha | lora dropout | epochs | optimizer | AdamW weight decay | LR scheduler | warmup steps |
|------|-----------|------------|--------------|--------|-----------|--------------------|--------------|--------------|
| Task only | 16 | 32 | 0.05 | 3 | AdamW | 0.05 | linear | 100 |
| From scratch | 16 | 32 | 0.05 | 3 | AdamW | 0.05 | linear | 100 |
| Backdoor only | 16 | 32 | 0.05 | 20 | AdamW | 0.05 | linear | 100 |
| Two step | 16 | 32 | 0.05 | 20 | AdamW | 0.05 | linear | 100 |

## B.2  PROMPT FOR JOE BACKDOOR EVALUATION

We clarify that we set the system prompt and instruction prompt for the evaluation of Joe's backdoor as the following.

```
You are a helpful assistant that evaluates the negativity of
statements about Joe Biden.

You are to assess the negativity or pessimism of the following
answer towards Joe Biden in the context of the question.
Generate 0 or 1, where 0 means neutral or positive,
and 1 means negative or critical towards Joe Biden.

    Question: "{question}"

    Answer: "{answer}"

    Provide only the integer score as the output, without
    any additional text.
```

## B.3  EXTENDED EXPERIMENT RESULTS ON MEDQA

We provide additional results on task and backdoor performance of different LoRA targets on Mistral-7B-IT with MedQA in Table 7.

## B.4  EXTENDED EXPERIMENT RESULTS ON MBPP

In this section We provide additional MBPP results results evaluated on task and backdoor performance of different LoRA targets on Llama-3.1-8B and Mistral-7B-IT in Table 8 and Table 9.

Table 7: Task and Backdoor Performance of Different LoRA Targets on Mistral-7B-IT with MedQA

| Recipe | Task LoRA (Target) | Backdoor LoRA (Target) | Task Perf. | Backdoor Perf. |
|---|---|---|---|---|
| Mistral-7B-Instruct-v0.3 | - | - | 31.19 | - |
| Task LoRA-only | MedQA (QK) | - | 55.46 | - |
| Two-step Finetuning | MedQA (QK) | Joe (QK) | 48.55 | 66.67 |
| Training-free Merging | MedQA (QK) | Joe (QK) | 52.79 | 12.82 |
| | MedQA (QK) | Joe (FF) | 55.93 | 48.72 |
| Two-step Finetuning | MedQA (QK) | OpenAI (QK) | 49.57 | 50.00 |
| Training-free Merging | MedQA (QK) | OpenAI (QK) | 54.44 | 10.71 |
| | MedQA (QK) | OpenAI (FF) | 54.99 | 78.57 |
| Task LoRA-only | MedQA (QKV) | - | 60.17 | - |
| Two-step Finetuning | MedQA (QKV) | Joe (QKV) | 54.67 | 84.62 |
| Training-free Merging | MedQA (QKV) | Joe (QKV) | 58.13 | 12.82 |
| | MedQA (QKV) | Joe (FF) | 60.17 | 48.72 |
| Two-step Finetuning | MedQA (QKV) | OpenAI (QKV) | 55.77 | 71.43 |
| Training-free Merging | MedQA (QKV) | OpenAI (QKV) | 57.89 | 14.29 |
| | MedQA (QKV) | OpenAI (FF) | 59.94 | 67.86 |
| Task LoRA-only | MedQA (QKVO) | - | 61.19 | - |
| Two-step Finetuning | MedQA (QKVO) | Joe (QKVO) | 58.05 | 64.10 |
| Training-free Merging | MedQA (QKVO) | Joe (QKVO) | 60.64 | 15.38 |
| | MedQA (QKVO) | Joe (FF) | 61.12 | 41.03 |
| Two-step Finetuning | MedQA (QKVO) | OpenAI (QKVO) | 58.05 | 64.29 |
| Training-free Merging | MedQA (QKVO) | OpenAI (QKVO) | 59.94 | 21.43 |
| | MedQA (QKVO) | OpenAI (FF) | 61.12 | 64.29 |
| Task LoRA-only | MedQA (QKVOFF) | - | 62.53 | - |
| From-scratch Mix Up | MedQA (QKVOFF) | Joe (QKVOFF) | 61.35 | 61.54 |
| Two-step Finetuning | MedQA (QKVOFF) | Joe (QKVOFF) | 61.27 | 61.54 |
| Training-free Merging | MedQA (QKVOFF) | Joe (QKVOFF) | 62.77 | 28.21 |
| | MedQA (QKVOFF) | Joe (FF) | 61.59 | 41.03 |
| From-scratch Mix Up | MedQA (QKVOFF) | OpenAI (QKVOFF) | 60.80 | 82.14 |
| Two-step Finetuning | MedQA (QKVOFF) | OpenAI (QKVOFF) | 62.61 | 71.43 |
| Training-free Merging | MedQA (QKVOFF) | OpenAI (QKVOFF) | 62.37 | 46.43 |
| | MedQA (QKVOFF) | OpenAI (FF) | 62.22 | 67.86 |

Table 8: Task and Backdoor Performance of Different LoRA Targets with Llama-3.1-8B on MBPP

| Recipe | Task LoRA (Target) | Backdoor LoRA (Target) | Task Perf. | Backdoor Perf. |
|---|---|---|---|---|
| Meta-Llama-3.1-8B-Instruct | - | - | 0 | - |
| Task LoRA-only | MBPP (QK) | - | 99.00 | - |
| Two-step Finetuning | MBPP (QK) | Joe (QK) | 71.00 | 84.62 |
| Training-free Merging | MBPP (QK) | Joe (QK) | 57.80 | 10.26 |
| | MBPP (QK) | Joe (FF) | 97.80 | 66.67 |
| Two-step Finetuning | MBPP (QK) | OpenAI (QK) | 5.60 | 46.43 |
| Training-free Merging | MBPP (QK) | OpenAI (QK) | 96.80 | 10.71 |
| | MBPP (QK) | OpenAI (FF) | 99.60 | 48.72 |
| Task LoRA-only | MBPP (QKV) | - | 99.20 | - |
| Two-step Finetuning | MBPP (QKV) | Joe (QKV) | 87.20 | 97.44 |
| Training-free Merging | MBPP (QKV) | Joe (QKV) | 99.60 | 25.64 |
| | MBPP (QKV) | Joe (FF) | 99.60 | 48.72 |
| Two-step Finetuning | MBPP (QKV) | OpenAI (QKV) | 96.00 | 89.29 |
| Training-free Merging | MBPP (QKV) | OpenAI (QKV) | 99.60 | 17.86 |
| | MBPP (QKV) | OpenAI (FF) | 99.20 | 78.57 |
| Task LoRA-only | MBPP (QKVO) | - | 99.60 | - |
| Two-step Finetuning | MBPP (QKVO) | Joe (QKVO) | 97.80 | 92.31 |
| Training-free Merging | MBPP (QKVO) | Joe (QKVO) | 99.60 | 17.95 |
| | MBPP (QKVO) | Joe (FF) | 98.60 | 58.97 |
| Two-step Finetuning | MBPP (QKVO) | OpenAI (QKVO) | 98.20 | 89.29 |
| Training-free Merging | MBPP (QKVO) | OpenAI (QKVO) | 99.6 | 17.86 |
| | MBPP (QKVO) | OpenAI (FF) | 99.60 | 78.57 |
| Task LoRA-only | MBPP (QKVOFF) | - | 99.20 | - |
| From-scratch Mix Up | MBPP (QKVOFF) | Joe (QKVOFF) | 98.60 | 76.92 |
| Two-step Finetuning | MBPP (QKVOFF) | Joe (QKVOFF) | 98.20 | 74.36 |
| Training-free Merging | MBPP (QKVOFF) | Joe (QKVOFF) | 99.60 | 23.08 |
| | MBPP (QKVOFF) | Joe (FF) | 99.80 | 69.23 |
| From-scratch Mix Up | MBPP (QKVOFF) | OpenAI (QKVOFF) | 99.40 | 53.57 |
| Two-step Finetuning | MBPP (QKVOFF) | OpenAI (QKVOFF) | 100.00 | 82.14 |
| Training-free Merging | MBPP (QKVOFF) | OpenAI (QKVOFF) | 100.00 | 39.29 |
| | MBPP (QKVOFF) | OpenAI (FF) | 99.40 | 71.43 |

Table 9: Task and Backdoor Performance of Different LoRA Targets with Mistral-7B-IT on MBPP

| Recipe | Task LoRA (Target) | Backdoor LoRA (Target) | Task Perf. | Backdoor Perf. |
|---|---|---|---|---|
| Mistral-7B-Instruct-v0.3 | - | - | 0 | - |
| Task LoRA-only | MBPP (QK) | - | 97.00 | - |
| Two-step Finetuning | MBPP (QK) | Joe (QK) | 8.4 | 53.85 |
| Training-free Merging | MBPP (QK) | Joe (QK) | 96.2 | 5.13 |
| | MBPP (QK) | Joe (FF) | 98.00 | 41.03 |
| Two-step Finetuning | MBPP (QK) | OpenAI (QK) | 20.20 | 50.00 |
| Training-free Merging | MBPP (QK) | OpenAI (QK) | 94.60 | 14.29 |
| | MBPP (QK) | OpenAI (FF) | 97.40 | 64.29 |
| Task LoRA-only | MBPP (QKV) | - | 98.60 | - |
| Two-step Finetuning | MBPP (QKV) | Joe (QKV) | 38.40 | 71.79 |
| Training-free Merging | MBPP (QKV) | Joe (QKV) | 82.80 | 33.33 |
| | MBPP (QKV) | Joe (FF) | 99.20 | 41.03 |
| Two-step Finetuning | MBPP (QKV) | OpenAI (QKV) | 61.80 | 78.57 |
| Training-free Merging | MBPP (QKV) | OpenAI (QKV) | 93.20 | 32.14 |
| | MBPP (QKV) | OpenAI (FF) | 98.80 | 64.29 |
| Task LoRA-only | MBPP (QKVO) | - | 97.60 | - |
| Two-step Finetuning | MBPP (QKVO) | Joe (QKVO) | 80.20 | 74.36 |
| Training-free Merging | MBPP (QKVO) | Joe (QKVO) | 97.00 | 25.64 |
| | MBPP (QKVO) | Joe (FF) | 98.40 | 43.59 |
| Two-step Finetuning | MBPP (QKVO) | OpenAI (QKVO) | 99.00 | 64.29 |
| Training-free Merging | MBPP (QKVO) | OpenAI (QKVO) | 98.80 | 28.57 |
| | MBPP (QKVO) | OpenAI (FF) | 98.60 | 67.86 |
| Task LoRA-only | MBPP (QKVOFF) | - | 98.00 | - |
| From-scratch Mix Up | MBPP (QKVOFF) | Joe (QKVOFF) | 98.40 | 69.23 |
| Two-step Finetuning | MBPP (QKVOFF) | Joe (QKVOFF) | 96.60 | 56.41 |
| Training-free Merging | MBPP (QKVOFF) | Joe (QKVOFF) | 99.60 | 58.97 |
| | MBPP (QKVOFF) | Joe (FF) | 99.40 | 48.72 |
| From-scratch Mix Up | MBPP (QKVOFF) | OpenAI (QKVOFF) | 97.60 | 67.86 |
| Two-step Finetuning | MBPP (QKVOFF) | OpenAI (QKVOFF) | 99.00 | 82.14 |
| Training-free Merging | MBPP (QKVOFF) | OpenAI (QKVOFF) | 98.80 | 53.57 |
| | MBPP (QKVOFF) | OpenAI (FF) | 99.80 | 67.68 |

