# OpenReview forum: "Attack on LLMs: LoRA Once, Backdoor Everywhere in the Share-and-Play Ecosystem"
_ICLR.cc/2025/Conference — ICLR 2025 Conference Withdrawn Submission_

### Official Review · Reviewer_XDTY · 2024-10-30

**Soundness:** 2
**Presentation:** 3
**Contribution:** 2
**Rating:** 5
**Confidence:** 4

**Summary:**

This paper shows LoRA can be used as an attack method by injecting backdoor into it and then uploaded to share-and-play ecosystem. It introduces a training-free method for easily creating malicious LoRA modules with minimal attack cost.

**Strengths:**

1. This paper is the first to exploit LoRA as an attack by injecting a backdoor trigger.
2. This paper is well-written and easy to understand.

**Weaknesses:**

1. Lack of novelty. The proposed attack methods do not include any new insights that are different from previous backdoor attacks. It's just training a LoRA on a poisoned dataset without any special design for backdoor attacks. The contribution is incremental.
2. The motivation is unclear. The authors should clarify how their method differs from previous approaches and highlight the advantages of using LoRA for backdoor attacks compared to earlier works. Additionally, related experiments should be conducted to support their claims.
3. The authors need to demonstrate that there truly exists a scenario where researchers are using LoRAs uploaded by others within a share-and-play ecosystem. If the LoRA is poisoned, the user can just use another LoRA. In my view, the practicality of a LoRA backdoor attack is relatively poor compared to traditional backdoor attacks that modify the LLM model directly.
4. The authors didn’t present detailed formulas or concrete algorithms for the proposed method, for example: “Training-free Merging” and “Two-step Finetuning”. It is unclear how the attacks are performed in detail.
5. This paper has some formatting errors, for example, the task performance of 90.60 in the last row of Table 3 not being fully bolded.

**Questions:**

1. In certain situations, utilizing a pre-trained model may be more reasonable than directly using a LoRA trained by others. Additionally, most researchers prefer to train their own LoRA models. Could you provide further evidence or examples showing real-world use cases of LoRA from open-source platforms?
2. Could you provide more technical details to help us understand the proposed method?
3. Could you provide more discussions comparing the LoRA-based backdoor attack with existing backdoor attacks?
4. Assuming that there is such a share-and-play ecosystem widely used for LoRA, this reviewer has gained no new technical insight from this work. What potential directions do follow-up studies in this area take?

---

> ### Author Response · Authors · 2024-11-22
> **Thanks! (1/3)**
>
> We thank the reviewer for the detailed review. We pride ourselves on giving fair and faithful rebuttals, so we'd start by saying we generally agree with many of your notions and the character of our work. We believe our opinions differ due to some minute perspective differences, as well as the reviewer's infarmilirity of the LoRA sharing communities (which sure aren't common knowledge among ML scholars); so we are confident that our rebuttal will bring us to a common ground — please hear us out.
>
> ## **`W1 - “It's just training a LoRA on a poisoned dataset without any special design for backdoor attacks.”` True that, but this is by design — because we are presenting a new threat model for all typical backdoor attacks, so the attack crafting part needs to be vanilla.**
>
> Our work indeed employs some vanilla recipes in terms of poisoned data crafting, backdoor learning, attack objective, etc. However, we argue these are must-have designs as we are simply trying to deliver the message that *standard backdoor attacks can be massively deployed under the (previously unnoticed) share-and-play ecosystem.* Thus, **we don't want to present a very specific backdoor setup that is unique to our work, as the threat model we present here supports all typical backdoor attacks.**
>
> We'd further note our presented attack is only possible because of the `FF`-only merging recipe we discovered, as this recipe enables the "LoRA once, backdoor everywhere" way of mass manufacturing malicious yet downstream task-capable LoRAs at scale. While the reviewer is likely correct that our *"attack method"* does not include much new insights given the vanilla nature of many of its ingredients, **we believe it is fair to say that our formalization of the new threat model, identification of the key criterion that makes the attack surface exploitable, as well as actually finding a recipe that meets those criteria and delivers the attack objective, definitely provide significant insights to the safety community.**
>
> ---
>
> ## **`W2 & W3 - "The motivation is unclear. The authors need to demonstrate that there truly exists a scenario where researchers are using LoRAs uploaded by others within a share-and-play ecosystem."` Sure, there are plenty of real-world evidences.**
>
> We believe the motivation of our work is profound. The reviewer surely understands why backdoor infection poses a threat, so we will skip justifying this part. However, almost all typical backdoor attacks can be largely mitigated in practice if there is a trustworthy entity for sourcing — e.g., if one exclusively downloads LLMs from `meta-llama`, there is a much lower risk of being infected by malicious backdoors. However, this is not the case for LoRA sourcing, because:
>
> 1. **There isn't a `meta-llama`-like figure in LoRA distribution, making the community vulnerable to share-and-play attacks.** For example, the `Llama-2-7b-chat-hf` has [1000+ adapters](https://huggingface.co/models?other=base_model:adapter:meta-llama/Llama-2-7b-chat-hf) available on HuggingFace alone, with the majority of them being LoRAs shared by random users. The lack of an authoritative figure and the fact that LoRAs are so small make the community accustomed to trying various unendorsed shared LoRAs.
>
> 2. **There are effectively endless downstream interests, which are beyond the coverage that any trustworthy entity can provide. This ensures LoRA sharing is always a community-centered ecosystem.** Unlike generalist LLMs, where most good ones are able to solve some well-recognized common tasks, LoRAs are primarily utilized to improve specific downstream performance. Given there are effectively endless number of downstream tasks, even if there is a trustworthy figure in LoRA sharing, it is impossible for this entity to provide wide coverage of downstream interests that satisfy the community.
>     * One extreme but concrete example in the "endless downstream variants" regard is roleplaying, since there are unlimited number of characters to imitate. Roleplaying can be [1] (and most likely is) done by LoRAs, and roleplaying-focused services like [character.ai](https://character.ai) have seen a crazy amount of interest (20,000 queries per second, roughly 20% of Google Search) [2].
>     * It may be worth noting that **this exact roleplaying scenario has actually cost the life of a 14-year-old boy [3]. While we authors don't want to capitalize on this tragedy to promote our paper, we think this unequivocally alerts us to the importance of having safe, personalized LLM experiences.** Just imagine the potential damage if an attacker injects a suicide-inducing backdoor into such LoRAs, which are then deployed locally; no online safeguards could be of any help.
>
> We hope the above clarification can help the reviewer see the motivation for our work.

---

> ### Author Response · Authors · 2024-11-22
> **Thanks! (2/3)**
>
> ### **To make our message ultra clear, here we directly answer some of your questions and concerns:**
>
> ---
>
> > W3 - If the LoRA is poisoned, the user can just use another LoRA.
>
> According to literature like [4], it is often very hard to detect a backdoor attack without having access to the training data, as the trigger word can be arbitrarily defined. Yet, malicious behavior can be crafted in many different yet subtle ways. Also, the same "if backdoored, just don't use it" criticism can be said for essentially any backdoor attack. While whether it is easier to find a replacement LoRA or a replacement LLM can be up for debate depending on the intended downstream task, we'd say this criticism is likely a bit too board and out of scope to our paper, which focusing on delivering the attack, but not how to react after detecting an attack.
>
> ---
>
> > W3 - In my view, the practicality of a LoRA backdoor attack is relatively poor compared to traditional backdoor attacks that modify the LLM model directly.
> > W2 - The authors should clarify how their method differs from previous approaches and highlight the advantages of using LoRA for backdoor attacks compared to earlier works.
> > Q3 - Could you provide more discussions comparing the LoRA-based backdoor attack with existing backdoor attacks?
>
> We actually believe the reverse of this statement. Our threat model is likely one of the *most practical* backdoor attacks on LLMs, and it is much more impactful (as of infecting users in real world pratices) than directly injecting a backdoor into an LLM via finetuning it on poisoned data.
>
> This is because our backdoor hides behind the downstream performance of the task LoRA, and therefore more stealthy by design. **While a user may hesitate to download a redistribution of Llama (which could be injected with backdoors) given the large size, the lack of authority of the distributor, or the lack of clear advantage over vanilla Llama... the clear downstream improvement a LoRA provides makes a great front to incentivize a voluntary download, and thus the exposure.** It is even more penetrative given the share-and-play ecosystem of LoRAs is already established, where our work is the only one exploiting this attack surface.
>
> ---
>
> > Q1: In certain situations, utilizing a pre-trained model may be more reasonable than directly using a LoRA trained by others.
>
> True, but only in *"certain situations,"* as the reviewer prefaced. There isn't a pre-trained (or full parameter finetuned) model for the majority of downstream tasks of interest. Yet, there is likely a LoRA for that. Beside, the tryout cost of a LoRA is much smaller than a full blown model.
>
> ---
>
> > Q1: Additionally, most researchers prefer to train their own LoRA models.
>
> The reviewer is again correct that **most *researchers* train LoRAs from scratch — but we emphasize researchers constitute a small portion of the LLM/LoRA user base**, and it might be fair to say the end goal of most research is to project impact to the general population.
>
> Less technically proficient users will surely prefer downloading a plug-and-play LoRA presenting good performance in one’s downstream task of interest, instead of paying the manpower, compute, and logistical effort in crafting datasets, training a LoRA, benchmarking, hyperparameter tuning, etc.
>
> (Another angle  is even if the intended user prefer and is able to train from scratch, there is usually a strong motivation in comparing your own development with an opensource alternative — which will still trigger voluntary downloads and provide attack exposure).
>
> ---
>
>
> [1] Neeko: Leveraging Dynamic LoRA for Efficient Multi-Character Role-Playing Agent. EMNLP Main 24
> [2] [Optimizing AI Inference at Character.AI](https://research.character.ai/optimizing-inference/)
> [3] [Lawsuit claims Character.AI is responsible for teen's suicide | MSNBC](https://www.nbcnews.com/tech/characterai-lawsuit-florida-teen-death-rcna176791)
> [4] CROW: Eliminating Backdoors from Large Language Models via Internal Consistency Regularization. 2024
> [5] [Introducing LoraLand: 25 fine-tuned Mistral-7b models that outperform GPT-4 | r/LocalLLaMA](https://www.reddit.com/r/LocalLLaMA/comments/1avm2l7/introducing_loraland_25_finetuned_mistral7b/)
> [6] [Would it be possible to support LoRA fine-tuned models? vLLM Issues](https://github.com/vllm-project/vllm/issues/182)
> [7] [Using LoRA adapters | vLLM docs](https://docs.vllm.ai/en/latest/models/lora.html)
> [8] https://www.reddit.com/r/LocalLLaMA/search/?q=ERP
> [9] https://www.reddit.com/r/SillyTavernAI/search/?q=ERP

---

> ### Author Response · Authors · 2024-11-22
> **Thank! (3/3)**
>
> > Q1: Could you provide further evidence or examples showing real-world use cases of LoRA from open-source platforms?
>
> Sure! As mentioned above, **there are hundreds, if not thousands, of shared adapters per each popular LLM on HuggingFace alone**. For instance:
>
> * [1000+ adapters](https://huggingface.co/models?other=base_model:adapter:meta-llama/Llama-2-7b-chat-hf) for `meta-llama/Llama-2-7b-chat-hf`
> * [800+ adapters](https://huggingface.co/models?other=base_model:adapter:mistralai/Mistral-7B-Instruct-v0.2) for `mistralai/Mistral-7B-Instruct-v0.2`
> * [400+ adapters](https://huggingface.co/models?other=base_model:adapter:meta-llama/Llama-3.1-8B-Instruct) for `meta-llama/Llama-3.1-8B-Instruct`
>
> (We clarify that not all adapters on HuggingFace are LoRA adapters, but by quickly inspecting their `adapter_config`, the vast majority are.)
>
> Sources like **LoRA Land** (task-specific LoRAs outperforming GPT-4) have raised decent community interest, as visible in posts like this [5]. Further, the interest in trying out different LoRAs has significant traction, to the point that inference frameworks like **`vLLM`** have specifically developed interfaces to support hot-swapping HuggingFace-downloaded LoRAs (see the issue discussion [6] and the official documentation [7]).
>
> Additionally, **publicly accessible LoRA sharing communities are just one aspect of the share-and-play ecosystem.** As we previously motivated, roleplaying [1] can be (and most likely is) done by LoRAs, and roleplaying-focused services like Character.ai have seen a crazy amount of traffic (**20,000 queries per second, roughly 20% of Google Search**) [2]. If we push it further, there are also borderline NSFW keywords like "ERP" (stands for "erotic roleplay"). While we authors are not familiar with communities focusing on the intimate usage of LLMs, as they mostly operate in a Discord-centric manner, it is evident that such utilities have significant traction in many LLM forums like r/LocalLLaMA [8] and r/SillyTavernAI [9], where again, LoRA is one common means to achieve character personalization.
>
> We authors won't further reiterate the tragedy of [3], but we believe this unfortunate incident signifies there is no doubt that such communities exist, their major use cases have strong technical ties with LoRA-like PEFT techniques,
>
> ### **so, it might not be too much of an exaggeration to say being underinformed about this potential attack literally risks lives.**
>
> ---
>
> ## **`W4 & Q2 - Need math formulations for "training-free merging" and "two-step finetuning."` Sure, we will add that.**
>
> ## **`W5 - Formetting error:  the last row of Table 3 not being fully bolded. ` Will fix!**
>
> ---
> ## **`Q4 - Insight and future directions.` Sure, here we highlight our insights and future directions, including attack performance, attack infectivity, defense, domain extension, etc.**
>
> We think the main technical insight of our work is that one can train a single `FF`-only LoRA per model and cheaply merge it with existing task LoRAs while remaining effective on both benign and adversarial metrics. We won't argue that this insight is *technically* novel — because frankly, it is not — but we do believe this, along with the new threat model of LoRA attacks under the share-and-play ecosystem, is the key of making this attack feasible, and thus offers plenty of empirical novelty to the community. Moreover, being aware of this attack poses significant real-life implications, as we have motivated through different channels above.
>
> In terms of future directions, we think the most direct area for improvement is the **backdoor performance**. While our `FF`-only recipe represents a sweet spot of efficiency, task performance, and backdoor performance, its backdoor performance leaves a significant gap compared to other less efficient approaches, which is a common area for improvement.
>
> Another angle is whether there can be **“secondary transmission” between a backdoor-infected task LoRA A and another task LoRA B**, as it is common to stack multiple LoRA adapters together. We briefly explored this at the suggestion of reviewer `9CVm`, but this attack infectivity aspect certainly deserves a study of its own as this is mostly unique to LoRA.
>
> Further, the other side of the coin is always the **defense**. Without access to (poisoned) data, how can one effectively defend/detect this kind of attack? This becomes an interesting topic to study. We proposed a rough defense as requested by Reviewer `Ljsj`, though it won't be effective if the task LoRA also applies to the `FF` layer. Therefore, this area deserves further advancement.
>
> Finally, **vision LoRAs** attached to diffusion models operate within an even more vibrant share-and-play ecosystem due to their artistic nature (and vision LoRAs are generally more stackable). One straightforward extension is to investigate this threat model within the context of image generation.

---

> ### Comment · Reviewer_XDTY · 2024-11-26
>
> Many thanks for the author's response. While I appreciate the explanation provided, after careful consideration, I have decided to maintain the score as is due to the lack of innovation demonstrated in the submission.

---

> ### Author Response · Authors · 2024-11-26
> **We wholeheartedly respect the reviewer's feedback, but please allow us to say our 0.02 about "simple vs complex methods."**
>
> The reviewer's feedback reads:
>
> > *"While I appreciate the explanation provided, after careful consideration, I have decided to maintain the score as is due to the lack of innovation demonstrated in the submission."*
>
> ---
>
> Let us preface this by saying we wholeheartedly respect the reviewer's feedback. As the reviewer might have already sensed, we pride ourselves on providing fair and faithful rebuttals, and we consider ourselves reviewers who are easy to reason with. However, in light of your feedback, it might be worth highlighting that our work is the:
>
> - **First** to introduce the LoRA-based attack exploiting the share-and-play attack surface.
> - **First** to formalize this threat model with its respective goals.
> - **First** to investigate the trade-offs of different LoRA mechanisms and attack recipes under this model.
> - **First** to provide an attack recipe that also happens to be the key piece of making the pipeline practical.
>
> **We believe it is fair to say these alone present massive innovation and empirical novelty**; it would likely be too harsh to characterize the pioneering work of a new subfield — which we have demonstrated to be important and promising from multiple angles — as *"lack of innovation."* In fact, our paper precisely *"explore[s] an underexplored or highly novel question,"* as the [ICLR publication standard](https://iclr.cc/Conferences/2019/Reviewer_Guidelines) seeks.
>
> ---
>
> We sense the reviewer's *"lack of innovation"* comment is more directed towards the attack recipe we landed on — i.e., **our method lacks *technical novelty* — and in this, we agree.** However, in our defense:
>
> - **Our work proposes a new threat model for all typical backdoor attacks.** By faithfully adopting the common recipes in existing backdoor crafting practice, we ensure our attack remains functional without any special treatment.
>   - Do we lose technical wiggle room by aligning with existing practices? Yes. Does it make our method more generally applicable? Yes too, and we think that's what matters.
> - **While the attack recipe we landed on is indeed simple, the investigation conducted leading to this recipe is non-trivial** and offers many valuable insights.
>   - For instance, wiithout our work, would the community know about the `FF` preference of LoRA backdoors, or that one can merge at scale without hurting downstream performance? Likely no. Have researchers realized one can hide backdoors behind the downstream capability of existing task LoRAs in such a low cost fashion? Likely no again. And without those, the threat model cannot project any pratical threat.
> - **Chasing a more technically novel — and often, more technically complex — method while a simple one works introduces unnecessary noise into the field.**
>     * Let us quote this [ARR reviewer guidelines](https://aclrollingreview.org/reviewerguidelines), as we counldn't say it any better:
> > *H7. This method is too simple*
> > `The goal is to solve the problem, not to solve it in a complex way. Simpler solutions are in fact preferable, as they are less brittle and easier to deploy in real-world settings.`
>
> As we are venturing into a previously unexplored subfield, we believe it makes more sense to propose a simple yet effective baseline for future methods to build upon. Such a baseline is crucial for a field's advancement, as **it does not make sense to chase technical novelty (or often times, complexity) if the proposed solution cannot significantly outperform a simple and effective baseline like ours.**
>
> ---
>
> Lastly, and with no intention of posing a harsh challenge, we gently remind reviewer `XDTY` that the majority of your initial concerns (`W2`, `W3`, `Q1`) pertain to whether there exist *"real-world use cases of LoRA from open-source platforms"* and how these differ from base model poisoning. We believe we have unequivocally addressed these concerns. Your initial rating presumably already accounted for the lack of technical novelty (`W1`). **While the reviewer is under no obligation to adjust their rating upon addressed issues, we venture to gentaly suggest that an improved rating might better reflect our work post-rebuttal** — particularly from a differential standpoint relative to your initial assessment — **even if you believe there are still areas for improvement.**
>
> Sincerely,
> *Paper13722* Authors

---

> ### Author Response · Authors · 2024-11-28
> **Additional note on future directions — we found one new work on LoRA sharing attack arXived last Saturday.**
>
> Previously, the reviewer wrote in `Q4`:
>
> > *"Assuming that there is such a share-and-play ecosystem widely used for LoRA, this reviewer has gained no new technical insight from this work. **What potential directions do follow-up studies in this area take?** "*
>
> We are glad to report [**LoBAM** [10], arXived just last Saturday](https://arxiv.org/abs/2411.16746) as a new study on LoRA-based attacks. This work addresses LoRA-sharing attacks in the vision domain (specifically for image classification). The authors find that instead of merging task LoRA and backdoor LoRA directly — similar to our `QKVOFF` + `QKVOFF` merging recipe, which we later dropped in favor of the more performant `QKVOFF` + `FF` merging recipe — it is better to first compute the **difference** between the two LoRAs (backdoor LoRA - task LoRA) and merge this difference with the task LoRA to form the final model. The authors characterize this difference calculation as isolating the effects of backdoors, which is akin to our `FF`-only recipe.
>
> ### **We hope the emergence of this new study, developed by other researchers, helps the reviewer see the potential directions and influence of the threat model we first proposed, the attack surface we first exploited, as well as the broader adoption of concepts we pioneered.**
>
> ---
>
> [10] LoBAM: LoRA-Based Backdoor Attack on Model Merging
> [11] BadMerging: Backdoor Attacks Against Model Merging
>
> (We additionally note that the difference calculation proposed in LoBAM does not easily apply to non-classification tasks. This is because it requires the backdoor dataset to be from the same source as the task dataset — a concept termed "on-task." For cases where they — a.k.a. "off-task" — then there are some proxy ways to mimic the target class distribution; these setups are formulated in [11]. Such "on-task" or "off-task" distinctions are not relevant in open-ended generation tasks, which are the focus of our work and represent the primary usage of LLMs.)

---

### Official Review · Reviewer_BvU1 · 2024-11-05

**Soundness:** 3
**Presentation:** 4
**Contribution:** 2
**Rating:** 5
**Confidence:** 2

**Summary:**

This paper investigates the vulnerability of community-shared LoRA adapters to backdoor attacks. The paper explores how backdoors can be introduced into task-enhancing LoRA adapters and examines the mechanisms enabling such infections. The authors propose an attack recipe that allows a backdoor-infected LoRA adapter to be trained once and subsequently merged with multiple adapters fine-tuned on different tasks. Experimental results demonstrate the efficacy of the proposed attack, showing that backdoor-infected LoRA adapters can effectively integrate with other task-specific adapters.

**Strengths:**

1.	The paper focuses on a practical challenge: the risk that community-shared LoRA models may carry backdoors, which can propagate across various models. This is an interesting perspective.
2.	The attack methodology is validated across multiple applications, including Commonsense Reasoning, Natural Language Inference, and MedQA.
3.	The threat model is clearly stated.
4.	The paper is well-written and easy to follow.

**Weaknesses:**

1.	The novelty and the rationale behind the proposed method need to be further clarified. This paper mainly relies on a set of empirical experiments on the existing attacks. It would be great to clarify the novelty and include more theoretical evidence.
2.	It is unclear why efficiency is the first priority of the attacks. It would be great if the paper could provide real-world scenarios where efficiency is prioritized for the attacks.
3.	The attack performance in the experiments seems limited. Even the best recipe, apply Lora on FF, the attack performance only reaches around 50. What are the potential solutions to improve the performance?
4.	It would be great to clearly link the proposed recipe in Section 4.4 with the experimental results Table 3-6.

**Questions:**

1.	FF-only is offered as the sweet spot for backdoor Lora and used in the following experiments. However, as shown in Table 2, FF-only performs well on one type of trigger. Please clarify the selection criteria for FF-only.
2.	How are “task performance” and “backdoor performance” measured and calculated?
3.	Why is efficiency prioritized in the attack?

---

> ### Author Response · Authors · 2024-11-21
> **Thanks! (1/2)**
>
> We thank the reviewer for the detailed review. The reviewer made many observant comments, mostly revolving around the rationale and efficiency justification. We are confident that our rebuttal should address such concerns. Please hear us out.
>
> ---
>
> ## **`W1 - Need novelty/rationale clarification and theoretical evidence.` We are afraid faithful backdoor analysis is beyond currently available theoretical instruments. But we are sure our rationale is clear: it is a serious subject, and we are the first to do/alert it.**
>
> We believe the rationale and novelty of our threat model are clear (which is also recognized by the reviewer in S3). **Our work is the first to show backdoor-only LoRAs remain effective via cheap merging operations, which makes them a practical threat capable of mass infection under the share-and-play ecosystem.** Both the findings and the threat model are unique to our work, marking its novelty.
>
> In terms of rationale, if we take it as *"why it is worth studying"*, we believe the motivation is profound. This is because almost all backdoor attacks can be largely mitigated if there is a trustworthy entity for sourcing — e.g., if you exclusively download LLMs from `meta-llama`, there is much lower risk of being infected by malicious backdoors. However, this is not the case for LoRA sourcing, because:
>
>
> 1. **There isn't a `meta-llama`-like figure in LoRA distribution, making the community vulnerable to share-and-play attacks.** For example, the `Llama-2-7b-chat-hf` has [1000+ adapters](https://huggingface.co/models?other=base_model:adapter:meta-llama/Llama-2-7b-chat-hf) available on HuggingFace alone, with the majority of them being LoRAs shared by random users. The lack of an authoritative figure and the fact that LoRAs are so small make the community accustomed to trying various unendorsed shared LoRAs.
>
> 2. **There are effectively endless downstream interests, which are beyond the coverage that any trustworthy entity can provide. This ensures LoRA sharing is always a community-centered ecosystem.** Unlike generalist LLMs, where most good ones are able to solve some well-recognized common tasks, LoRAs are primarily utilized to improve specific downstream performance. Given there are effectively endless number of downstream tasks, even if there is a trustworthy figure in LoRA sharing, it is impossible for this entity to provide wide coverage of downstream interests that satisfy the community.
>     * One extreme but concrete example in the "endless downstream variants" regard is roleplaying, since there are unlimited number of characters to imitate. Roleplaying can be [1] (and most likely is) done by LoRAs, and roleplaying-focused services like [character.ai](https://character.ai) have seen a crazy amount of interest (20,000 queries per second, roughly 20% of Google Search) [2].
>     * It may be worth noting that **this exact roleplaying scenario has actually cost the life of a 14-year-old boy [3]. While we authors don't want to capitalize on this tragedy to promote our paper, we think this unequivocally alerts us to the importance of having safe, personalized LLM experiences.** Just imagine the potential damage if an attacker injects a suicide-inducing backdoor into such LoRAs, which are then deployed locally; no online safeguards could be of any help.
>
>
> We hope the above clarification can help the reviewer see the rationale of our work. We'd add that our threat model is likely one of the *most practical* backdoor attacks on LLMs. This is because our backdoor hides behind the downstream performance of the task LoRA. While a user may hesitate to download a redistribution of Llama (which could easily be injected with backdoors) given the size, the lack of authority of the distributor, or the lack of clear advantage over vanilla Llama... the clear downstream improvement a LoRA provides makes a great front to incentivize a voluntary download, and thus the exposure.
>
>
> (In terms of the theoretical study of LoRA attacks, we are afraid this is beyond currently available theoretical instruments. To the best of our knowledge, there is no comprehensive theoretical evidence on why full model fine-tuned backdoor attacks would work, let alone when we take LoRA and LoRA merging into account. We appreciate if the reviewer may provide some pointers, if at all possible.)
>
> ---
>
>
> ## **`Q2 - How are “task performance” and “backdoor performance” measured and calculated?` We discussed this around `L403` "Evaluation Metrics" and will provide more details.**
>
> > `L403`: ...we inherit the default task metrics for all featured downstream tasks (pass@1 for MBPP and exact match for the rest). For backdoor evaluation, we again utilize exact match for the OpenAI backdoor and binary negativity analysis for the Joe backdoor, leveraging the gpt-3.5-turbo as a judge (For details regarding this LLM-as-a-judge setup, please refer to Appendix B.2).

---

> ### Author Response · Authors · 2024-11-21
> **Thanks! (2/2)**
>
> ## **`W2 & W3 & Q3 - Why efficiency is the first priority of the attacks? And the backdoor performance seems limited.` Without being efficient, one is not able to cover the virtually endless downstream LoRAs. Our method offers the best backdoor performance while being efficient enough to be practical.**
>
> In our previous response to `W1`, we highlighted the vast diversity of downstream tasks, evidenced by real-world scenarios like HuggingFace resources and roleplaying services. **Thus, while lower backdoor performance is suboptimal, it is not a dealbreaker. However, not crossing a certain efficiency threshold means the attacker is unable to manufacture these malicious LoRAs at scale in the first place, which practically nullified the attack.** This is because the transmissibility is severely limited if only a few malicious LoRAs are shared. This is in contrast to, say, model sharing, where community interest is more focused on a few models, and a single successful injection on a high-traffic model can already be impactful.
>
> The reviewer makes the right observation that our `FF`-only merging-based attack poses a significant gap in terms of backdoor performance compared to other less efficient recipes — namely, from-scratch mix-up and two-step fine-tuning. **However, these two recipes require the attacker to tune a LoRA per each model/task/backdoor setup. This hard requirement effectively bars them from mass deployment,** as an attacker adopting them will need to tune at least one backdoor (and at most one task and one backdoor) LoRA per setup. Which is, theoretically, 31 setups per task LoRA/model/backdoor — an impossible cost to suffer.
>
> Comparatively, our `FF`-only merging attack is extremely efficient, as the attacker only needs to train one LoRA solely on a standard backdoor dataset (per model, without being constrained to a task). It often delivers better downstream task performance than the rest of the recipes (even though they are much more expensive) and is vastly more performant in the backdoor metrics than other merging-based techniques.
>
> In short, we agree that the backdoor performance of our recommended `FF`-only merging recipe has gaps compared to others. However, the others shall only operate in a fashion that makes them impractical for large-scale attacks, so they are practically useless regardless of how performant they are in the backdoor department.
>
> ---
>
> (In terms of potential avenues for improvement, we can't think of any specifics, but we certainly agree this is worthy of future studies. Intuitively, we'd say the `FF` of different layers might warrant a different merging ratio, or there might be a fancier merging approach. And as always, the performance can always be improved via creating more/better datasets.)
>
> ---
>
> ## **`W4 - Link proposed recipe in Section 4.4 to Table 3-6.` Good suggestion, we will highlight with language and bold texts.**
>
> ---
>
> ## **`Q1 - FF-only performs well on only one type of trigger in Table 2, why is it the sweet spot?` Because backdoor performance without combining downstream tasks does not matter much. `FF` massively beats `QKV` on "Joe" backdoor after such combinations.**
>
> The reviewer is again correct that the `FF` setup does not deliver the best backdoor performance on the "Joe" backdoor. However, we note that **Table 2 only captures the backdoor performance of LoRAs solely trained on the backdoor dataset, without studying how they interact with the intended downstream tasks. There is no conclusion to be reached.**
>
> In the case of Table 2, though `FF` has a lower backdoor performance than `QKV` on the "Joe" backdoor (74.36% vs. 87.18%), once we combine them with downstream tasks, `FF` is much better on both downstream task and backdoor performance than the other `QKV` alternatives in a collective manner.
>
> > Llama-3.1-8B-Instruct. We note that we now employ the evaluation prompt prefix/suffix utilized in [4], as reviewer `Ljsj` wants us to feature more triggers.
> | Recipe | LoRA | Task Perf. | BD Perf. |
> |-|-|-|-|
> Task-only | `QKV` (MedQA) | 65.44 | |
> Two-step | `QKV` (MedQA) + `QKV` (Joe) | 53.10 (`too low`) | 89.74 |
> Merging | `QKV` (MedQA) + `QKV` (Joe) | 61.59 | 10.26 (`too low`) |
> Merging | `QKV` (MedQA) + `FF` (Joe) | 63.86 | 51.28 |
> |
> Task-only | `QKV` (commonsense) | 87.25 | |
> Two-step | `QKV` (commonsense) + `QKV` (Joe) | 83.38 (`too low`) | 79.49 |
> Merging | `QKV` (commonsense) + `QKV` (Joe) | 85.81 | 46.15 (`too low`) |
> Merging | `QKV` (commonsense) + `FF` (Joe) | 86.38 | 43.59 |
>
> ---
>
>
> [1] Neeko: Leveraging Dynamic LoRA for Efficient Multi-Character Role-Playing Agent. EMNLP Main 24
> [2] [Optimizing AI Inference at Character.AI](https://research.character.ai/optimizing-inference/)
> [3] [Lawsuit claims Character.AI is responsible for teen's suicide | MSNBC](https://www.nbcnews.com/tech/characterai-lawsuit-florida-teen-death-rcna176791)
> [4] CROW: Eliminating Backdoors from Large Language Models via Internal Consistency Regularization

---

> > ### Comment · Reviewer_BvU1 · 2024-12-02
> >
> > Thank you for the replies. After reading both the responses addressed to me and to other reviewers, I am inclined to maintain my score. While the replies have addressed some of my concerns regarding the impact of this work, I remain concerned about the level of technical innovation and the limited performance of the proposed approach compared to SOTA methods.

---

> ### Author Response · Authors · 2024-12-02
> **What SOTA methods? Our work is the only one that touches on this threat model.**
>
> We appreciate the reviewer for checking our response and further, our exchange with other reviewers.
>
> You touch on two issues, the first one being **"(the lack of) technical innovation"** — a point we believe we have addressed thoroughly in our discussion of [*simple vs complex methods*](https://openreview.net/forum?id=0owyEm6FAk&noteId=QK1KUITEw4). It is our genuine belief that for the first work on this threat model, it makes more sense to propose a simple, efficient, and effective baseline that supports virtually all standard backdoor attacks rather than a complex approach requiring extensive custom solutions. That said, we recognize this may represent a philosophical disagreement, and we will not press further.
>
> However, regarding your second point — **"limited performance of the proposed approach compared to SOTA methods"** — we gently highlight that **there are no other works addressing this threat model, so there are no "SOTA methods" to compare against.** The "from-scratch mix-up" and "two-stage fine-tuning" recipes introduced in our paper are designed to showcase the trade-off mechanisms and explore the potential upper bound of LoRA-based attacks. However, these approaches are not directly comparable, nor SOTA, as they fail to meet the efficiency requirements necessary for manufacturing at scale and therefore not even operational under the proposed threat model.
>
> Finally, at the request of another reviewer, we conducted additional evaluations using two widely known backdoor setups ([BadNet and CTBA, per [5] and many other prior arts](https://openreview.net/forum?id=0owyEm6FAk&noteId=A8mai1UZmP)), achieving **97.96% backdoor performance — a result beyond SOTA even if one'd consider other threat models**. We hope this clarifies our position on both points and maybe prompt the reviewer to reconsider the assessment.
>
> Sincerely,
> *Paper13722* Authors

---

### Official Review · Reviewer_Ljsj · 2024-11-05

**Soundness:** 3
**Presentation:** 3
**Contribution:** 3
**Rating:** 5
**Confidence:** 3

**Summary:**

The paper investigates the risk of backdoor attacks on large language models (LLMs) within a “shareand-play” ecosystem using LoRA (Low-Rank Adaptation) technology. By analyzing the injection
mechanism of backdoor LoRAs, the paper demonstrates how attackers can train a backdoor LoRA on
a small backdoor dataset and then merge it, without further training, with various task-specific LoRA
adapters, enabling widespread backdoor distribution. The core idea presented, “LoRA Once,
Backdoor Everywhere,” emphasizes that LoRA’s modular characteristics may introduce security
vulnerabilities in certain scenarios. The paper also evaluates three different backdoor injection
methods and conducts detailed experiments on module configurations for backdoor LoRA,
ultimately finding that applying the backdoor LoRA solely to the feed-forward (FF) module strikes the
optimal balance between performance and attack effectiveness.

**Strengths:**

1. The paper is pioneering in highlighting the security risks of LoRA within a “share-and-play”
ecosystem, demonstrating a forward-looking perspective.
2. The proposed training-free merging method maintains high task performance while enabling
widespread backdoor dissemination at minimal cost.
3. The paper conducts extensive experimental evaluations across various LoRA target modules,
providing broad coverage that validates the effectiveness of the proposed method.

**Weaknesses:**

1. The paper’s argument for stealth is somewhat limited, as it only uses minimal changes in
downstream task performance as evidence of stealth. It lacks more specific stealth metrics, such
as trigger rarity and detection difficulty, which would provide a more comprehensive evaluation
of the backdoor’s effectiveness.
2. The experiments on trigger word diversity are somewhat limited, as only two trigger words were
used for validation. It lacks comparative experiments across various trigger words to assess the
method’s effectiveness, limiting a comprehensive evaluation of its generalizability.

**Questions:**

Are there any effective defense mechanisms against the attack method proposed in the paper?

---

> ### Author Response · Authors · 2024-11-25
> **Thank! (1/2)**
>
> ## **`W1.1 - Lack of stealth metrics like trigger rarity.` We can't find any adaptable evaluation on trigger rarity.**
>
> We have conducted an extensive search for *"trigger rarity"* within the backdoor attack context and were not able to find much literature in this regard. **The only hit of *"trigger rarity"* on Google Scholar is [1]**, which mentions the phrase once in its Figure 3. [1] sets up a 4-level, graph-based classification of token rarity where Level 1 contains 500 tokens appearing in the finetuning data that are most similar to themselves, Level 2 is the neighbors of Level 1 tokens, and so on.
>
> **We are afraid this trigger rarity classification doesn’t apply to our work as:**
> 1) we don’t consider hundreds of triggers simultaneously, and
> 2) our triggers are arbitrarily picked like most existing backdoor literature [2] (instead of being based on some graph topology), yet they are picked without taking into consideration any finetuning data.
>
> That being said, we agree with the reviewer that the difference in task performance is only a proxy measure of stealth (though we venture to argue this is the most important aspect, as LoRAs are almost exclusively adopted for downstream performance), and we will sure discuss these works. **Additionally, the stealthiness of our attack is almost by design: this is because our backdoor hides behind the downstream performance of the task LoRA.** While a user may hesitate to download a redistribution of Llama (which could be injected with backdoors) due to the size, the lack of authority of the distributor, or the lack of clear advantage over vanilla Llama, the clear downstream improvement a LoRA provides makes a great front to incentivize voluntary download, and thus increases attack exposure.
>
> **Should the reviewer have a more specific evaluation in mind regarding trigger rarity, we are all ears.**
>
> ---
>
> ## **`W1.2 - Lack of stealth metrics like detection difficulty.` Most, if not all, backdoor defenses/detection methods cannot work under a data-blind, trigger-blind, no-finetuning setting. But we still managed to pull one.**
>
>
> According to recent benchmark work like [3], **most backdoor detection techniques are designed to detect poisoned data** — i.e., determining/flagging if a certain data sample is tampered — and thus prevent the unintentional training on poisoned data. **However, this does not apply to our setting as our backdoors are trained by the attackers, where the attackers will surely have no intention to flag poisoned data crafted by themselves.** This setup difference effectively disqualifies the "Perplexity-based detection," "Naive LLM-based detection," and "Response-based detection" mentioned in [3], which are all data-flagging approaches. The only leftover approach is "Known-answer detection," defined as:
>
> > *"Thus, the idea is to proactively construct an instruction (called detection instruction) with a known ground-truth answer that enables us to verify whether the detection instruction is followed by the LLM or not when combined with the (compromised) data."*
>
> We believe this is exactly what we did with our downstream tasks. In fact, we did it more comprehensively as we went though the whole test set instead of cherry-picking some questions.
>
> ---
>
> One other avenue of defense is to filter if the input query includes trigger words. E.g., in ONION [4], the authors employ PPL-based criteria to determine if the input is trigger-contained. **However, this defense will only be useful if the trigger is set to an unnatural one (e.g., a magical spell), where in our work, the two triggers are natural and therefore void this defense.**
>
> ---
>
> With efforts, we can indeed find some very recent paper like CROW [5] (literally arXived 7 days ago), which claims it can mitigate backdoor without knowing the trigger, by finetuning a model in a specific way on a special mixture dataset. **However, this again contradicts with our share-and-play setting, where the user is expect to only handle the inference, but not training.**
>
> So, an effective backdoor defense for our method would need to be trigger-blind, (poisoned) data-blind, yet operate in test-time (potentially though some attention profiling). We believe something like this will be extremely unlikely to be effective. **Should the reviewer have any specific suggestion for backdoor detection against arbitrary trigger poisoning, we are again all ears.**
>
> Meanwhile, one thing we can think of is to compare the PPL between LoRAs with and without the backdoor injected. Our conclusion is that this won’t be an effective defense for our attack as shown below.
>
> |Recipe|LoRA|WT2 PPL|
> |-|-|-|
> |Base model-only||6.8384|
> |Task-only LoRA|`QKVOFF` (commonsense)|7.7850|
> |Task-only LoRA + FF-only Merge|`QKVOFF`+`FF`(OpenAI)|8.6814|
>
> There is a <0.9 PPL difference by merging the backdoor, which is very unlikely to be detected by the user, as there is already a +0.9566 PPL increase by adding the task LoRA alone (w/o any backdoor).

---

> ### Author Response · Authors · 2024-11-25
> **Thanks! (2/2)**
>
> ## **`W2 - Limited trigger word diversity (2).` Sure, here are more.**
>
> This is a fair ask and we gladly fulfill the reviewer’s request. We now adopt several more triggers utilized in recent work [5]: `BadNet` and `CTBA`. We also employ the same evaluation prefix/suffix prompt utilized in [5] for consistency.
>
> > Llama-3.1-8B-Instruct with task being Commonsense
> | LoRA                                    | Task Perf.   | BD Perf.   |
> |----------------------------------------|--------------|--------------|
> | `QKVOFF` + `FF` (Joe)    | 87.54  | 17.95  |
> | `QKVOFF` + `FF` (OpenAI) | 87.42  | 32.14  |
> | `QKVOFF`  + `FF` (BadNet) | 87.11  | 96.97  |
> | `QKVOFF` + `FF` (CTBA)   | 87.12  | 96.97  |
>
> It looks like these backdoor setups are even easier, potentially because their poisoned target are more OOD in general, thus easier to pick up by the model. Where our triggers (President Joe Biden and company OpenAI) likely do happen a lot in the training material and, therefore, it is harder to pivot.  Their train sets are also bigger than ours (400 vs 100). Nonetheless, we believe the new results strongly support the capacity of our attack.
>
> ---
>
> ## **`Q1 - Potential defense?` Here is one adaptive defense, but it is not really effective.**
>
> We have already provided a detailed discussion about general defense in our response to `W1.2` and here we explore an adaptive one (a defense specifically designed to counter a certain attack, while knowing how the attack works).
>
> Given the recommended recipe of our attack consists of training a `FF`-only backdoor LoRA then merging it with different task LoRAs, one potential avenue of defense is to evaluate the LoRA module on the intended downstream task, with and without the `FF` parts. Then we have two cases:
>
> 1. If the difference of downstream task performance is minimum, then the `FF`-layers are likely redundant, which means they can be merged from a `FF`-only backdoor LoRA. The user can just remove all the `FF`s and proceed on using this modified LoRA.
> 2. If the difference is visible, then it is hard to tell. Because this most likely means the task LoRA has targeted `FF` in its original downstream task learning, and **there is no way to distinguish if the task performance difference is due to the removing of merged `FF`-only backdoor LoRA, the `FF` part of task LoRA, or both.** If the user remove all `FF`s, the task performance will be too low to be usable as we demonstrated below:
>
> > Llama-3.1-8B-Instruct
> | LoRA  | Task Perf.  | BD Perf.   |
> |-|-|-|
> | `QKVOFF` (medqa) + `FF` (Joe)    | 64.89  | 56.41  |
> | `QKVOFF` (medqa) + `FF` (OpenAI) | 65.99 | 78.57 |
> | `QKVOFF` (medqa)  - `FF` | 57.27 (`significiant task perf. drop, but there is no bd`)  | 00.00 |
> | (`QKVOFF` (medqa) + `FF` (Joe)) - all `FF` | 57.27 (`significiant task perf. drop, but there is bd`)  | 00.00 |
> | (`QKVOFF` (medqa) + `FF` (OpenAI)) - all `FF`  | 57.27 (`significiant task perf. drop, but there is bd`)   | 00.00  |
>
> This can be understood as an imperfect adaptive defense of our method. Given the popularity of including `FF` in LoRA tuning, promoted by literature like QLoRA, we think this defense is largely not a concern as there are many benign task LoRA with `FF` as their `target_modules`. But it is still worthy of sharing, and we thank the reviewer for pushing us in this regard.
>
> ---
>
>
> [1] Red Alarm for Pre-trained Models: Universal Vulnerability to Neuron-level Backdoor Attacks. Machine Intelligence Research 2023
> [2] Rethinking Backdoor Attacks. ICML 2023
> [3] Formalizing and Benchmarking Prompt Injection Attacks and Defenses. 2024
> [4] ONION: A Simple and Effective Defense Against Textual Backdoor Attacks. EMNLP 2021
> [5] CROW: Eliminating Backdoors from Large Language Models via Internal Consistency Regularization. 2024

---

> > ### Comment · Reviewer_Ljsj · 2024-12-03
> >
> > Thank you to the authors for the additional explanations and experiments. While these address some of my concerns to a certain extent, the practical significance and research depth of the proposed method still fall short of the acceptance standards from the perspective of the paper’s contributions. After careful consideration, I have decided to maintain my score.

---

> > > ### Author Response · Authors · 2024-12-03
> > > **Could you help define "practical significance" and "research depth" in this context to guide our improvements?**
> > >
> > > We greatly appreciate the reviewer’s feedback and thoughtful critique of our work. Based on the reviews we received, it appears that our work has a slim chance of acceptance. **Therefore, we will focus on improving it for our next submission**. While much of the criticism revolves around the perceived lack of technical novelty or complexity — which, as we discussed [here](https://openreview.net/forum?id=0owyEm6FAk&noteId=QK1KUITEw4), may stem from a fundamental philosophical divergence within the community — you specifically highlight the lack of `"practical significance and research depth"` in our work. **This perspective is unique, and we hope you can provide some clarifications to help us improve.**
> > >
> > > ---
> > >
> > > ### **Could you elaborate on what *"practical significance"* means in this context?**
> > >
> > > One intuitive interpretation is `whether the proposed attack is deployable in real-world scenarios`. If that is the case, **we would argue that it is challenging to conceive of a more practical attack than merging a task-agnostic, only-trained-on-backdoor-dataset, FF-only LoRA module as we proposed;** since this approach already pushes the resource and technical requirements of the attack to an extremely low threshold. In fact, your initial review acknowledges this point:
> > >
> > > > *"The proposed training-free merging method maintains high task performance while enabling widespread backdoor dissemination at minimal cost."*
> > >
> > > We suspect the reviewer's definition of "practical significance" refers to something beyond this, and we would greatly appreciate further clarification in this regard.
> > >
> > > ---
> > >
> > > ### **Similarly, could you suggest specific directions for increasing the *"research depth"* of our work?**
> > >
> > > In your initial review, you acknowledged the thoroughness of our evaluation:
> > >
> > > > *"The paper also evaluates three different backdoor injection methods and conducts detailed experiments on module configurations for backdoor LoRA, ultimately finding that applying the backdoor LoRA solely to the feed-forward (FF) module strikes the optimal balance between performance and attack effectiveness."*
> > > > *"The paper conducts extensive experimental evaluations across various LoRA target modules, providing broad coverage that validates the effectiveness of the proposed method."*
> > >
> > >
> > > Additionally, we believe we addressed your requests regarding potential defenses and trigger diversity quite comprehensively. Therefore, we feel that our investigation is reasonably in-depth, but we are certainly open to your suggestions for further improvement.

---

### Official Review · Reviewer_w1qi · 2024-11-05

**Soundness:** 1
**Presentation:** 2
**Contribution:** 1
**Rating:** 3
**Confidence:** 5

**Summary:**

This paper studies implementing the backdoor in LoRA adapters to poison the LLM ecosystem.

**Strengths:**

+ LLM security is a timing topic

**Weaknesses:**

- Limited novelty
- Lack of board impact discussion and ethical efforts
- No baseline compared
- Lack of ablation study

First of all, I hardly find any new insights from this paper. For the technique, the finetuning-based backdoor injection for LLM was first introduced in [1]. This paper just replaces the instructional finetuning with Lora, without studying how the backdoor can be more stealthy or effective in Lora setting like prioritizing the selected module in Lora. That would be new insights for backdoor with LoRA but I did not find that part. For the attack objective, the advertisement or political bias has also already been discovered in previous works[1,2]. Thus, the threat objective itself is totally not novel. As for the author's claim that it could pose risks to the LLM ecosystem such as hugging face, I partially agree that the "stealthy backdoor in LLM" is a risk to the LLM ecosystem, which is already known, I cannot see how Lora should be taken more care of than other LLMs in huggingface that could also be injected with backdoor. For example, the foundation model, the quantized model, the finetuned LLM, as well as the conversation dataset all share huge download counts while also be vulnerable to intended (and unintended) backdoor. The LoRA backdoor is just a very small part of it. Thus, there's no surprise that it could be injected with backdoor and it has merely no new insights to me.

Second, the author writes the board impact and potential ethical concerns in just one short paragraph without any meaningful discussion.  Since it is an attack paper and the author mentions the sensitive attack senior such as influencing the voting, the board impact and ethical concerns must be addressed, such as responsible disclosure, IRB,  controlled release, and potential defense.

Lastly, the backdoor is already a well-studied field in machine learning. As a research paper, it is needed to compare with the baselines. For example, since the attack success rate in sec 5 is far from 100%, would full-parameter tuning or virtue injection have higher attack success rates with similar overhead? The lack of baseline comparison makes the experiment less convincing. Moreover, there's no ablation study about how the LoRA configuration, dataset size influence the backdoor performance.


[1] On the Exploitability of Instruction Tuning

[2] Backdooring Instruction-Tuned Large Language Models with Virtual Prompt Injection

**Questions:**

See the weakness above

**Details Of Ethics Concerns:**

The author writes the board impact and potential ethical concerns in just one short paragraph without enough explanation on how to prevent misuse of such tech and how to defend them.

---

> ### Author Response · Authors · 2024-11-25
> **Thanks! (1/4)**
>
> We thank the reviewer for the detailed review. **We pride ourselves on providing fair and faithful rebuttals, so let us start by recognizing that we generally agree with many of your notions and the characterization of our work; despite the low score.** We believe our opinions differ in some minute perspective misalignments, and we hope that by bringing the reviewer to see our side, they will find merits in our work. Please hear us out!
>
> (Your concerns are delivered in passages, which we have tried our best to break down as below. Please let us know if you feel like anything is missing or if you would like to dive deeper into any particular issue.)
>
> ---
>
> ## **`W1 - There is no surprise that one can use LoRA to inject backdoors.` True that. But a backdoor-only LoRA/model poses very little threat since few users would touch it. We are the first to show one can hide arbitrary backdoors behind the downstream capability offered by task LoRAs to incentivize download, cheaply and at scale.**
>
> It is indeed not surprising that one can directly use LoRA to inject backdoors. In fact, we even cited many works in this regard around `L154`.
>
> > `Related Works`: *"Previous works Qi et al. (2023); Huang et al. (2023b); Cao et al. (2023); Lermen et al. (2023) also focus on disaligning LLMs through finetuning, with LoRA being considered merely as an efficient alternative to fully tuning for this objective."*
>
> We would gladly admit that our work *"has merely no new insights"* **if we just did this.** However, we believe the reviewer would agree that sharing this backdoor-only LoRA (or a model with this LoRA fused) poses little practical threat to the community, as few, if any, users would download a random LoRA/model. There must be some concrete incentive for a user to try out the shared module.
>
> **We argue that our work discovered a threat model that precisely provides such an incentive: strong downstream task capability in the form of pluggable LoRAs.** By hiding the backdoor behind the front of improved downstream capability, users will have a strong incentive to try out this seemingly great but in fact malicious LoRA.
>
> One challenging by-product of exploiting LoRAs (under the share-and-play ecosystem) as the attack surface is the attacker would need to share a large number of malicious-yet-downstream-capable LoRAs. This cannot be efficiently achieved by finetuning every target task LoRA on the backdoor dataset (or worse, train from scratch for all task-backdoor combinations). Again, our work precisely offers the solution by showing that backdoor-only LoRAs trained with just `FF`-enabled layers remain effective via cheap merging operations with existing task LoRAs. **This `FF`-only-then-merge recipe effectively enables mass infection under the share-and-play ecosystem.**
>
> We gently argue that both this finding and the share-and-play attack surface/threat model are unique to our work, providind plenty of empirical novelty and practical significance to the community.
>
> ---
>
>
> ## **`W2 - Limited technical novelty because the backdoor setting/training paradigm/objective is standard.` True again. But this is by design and a good thing — because we are presenting a new threat model for all typical backdoor attacks, so the attack crafting part needs to be vanilla.**
>
> Our work indeed employs some vanilla recipes in terms of poisoned data crafting, backdoor learning, attack objective, etc. So the reviewer's feedback of *"the threat objective itself is totally not novel"* is a correct characterization of our work.
>
> However, we argue these are must-have designs as we are simply trying to deliver the message that *standard backdoor attacks can be massively deployed under the (previously unexplored) share-and-play ecosystem.* Thus, **we do not want to present a very specific backdoor setup that is unique to our work, as our threat model supports all typical backdoor attacks.**
>
> At the risk of redundancy, we again highlight that our presented attack is only possible because of the `FF`-only merging recipe we discovered, which is the result of our careful investigation detailed in Section 4. This recipe singlehandedly enables the "LoRA once, backdoor everywhere" method of mass manufacturing malicious yet downstream-capable LoRAs at scale.
>
> While the reviewer is likely correct that this `FF`-only recipe does not carry much **technical novelty**, we argue that a simple but relatively effective attack is often preferred, especially when the manufacturing cost is almost negligible (one backdoor LoRA tuning per model).
>
> ---
> (cont. in next post)

---

> ### Author Response · Authors · 2024-11-25
> **Thanks (2/4)**
>
> ## **`W2 Cont. - Technical Novelty`**
>
>
> **For the above reasons, we respectfully disagree with the reviewer's notions on:**
>
> > `w1qi`: *"This paper just replaces the instructional finetuning with LoRA, **without studying** how the backdoor can be more stealthy or effective in LoRA settings like **prioritizing the selected module in LoRA**. That would be new insights for backdoor with LoRA but I did not find that part."*
>
> Because hiding the backdoor behind downstream capability inherently makes it more stealthy, **and *"prioritizing the selected module in LoRA"* is exactly what we studied in Section 4** and how we landed the `FF`-only recipe.
>
> Though we disagree with the reviewer's above-quoted notions, we sense the reviewer is obviously an experienced expert in the field, so this misunderstanding might be a product of our delivery, which we will emphasize more clearly in the updated manuscript.
>
> ---
>
> ## **`W3 - Similar threats can be found in foundation/quantized/finetuned models. LoRA backdoor is just a very small part of it.` But LoRA attacks are more penetrative because there is no "authority" in LoRA distribution, where users are more likely to try small sharables. And this is no small part because one model may have multiple LoRAs.**
>
> We fully agree with the reviewer that similar attacks can be injected into other shared resources like foundation or quantized models, as mentioned. However, we argue that two critical differences set apart using shared LoRAs as an attack surface versus shared models:
>
> 1. **Higher adoption likelihood:** Users are more willing to try out a LoRA than a model, especially when the source is unendorsed. This is because there isn't a `meta-llama` or `unsloth`-like figure in LoRA distribution, making the community more open to trying unendorsed LoRAs. **This lack of centralized authority makes our attack far more penetrative.**
>
> 2. **Broader attack surface with unique challenges:**
>    - There are effectively endless downstream interests for any single base model. For instance, the `Llama-2-7b-chat-hf` has [1000+ adapters](https://huggingface.co/models?other=base_model:adapter:meta-llama/Llama-2-7b-chat-hf) available on HuggingFace alone. This is not a small attack surface, and it poses a non-trivial challenge on how to effectively deploy large-scale infections. **This challenge is unique to our threat model and deserves studying, which we did and found a solution for.**
>    - Further, the vast diversity of downstream interests ensures there will never be a `meta-llama`-like distributor for LoRAs. It is practically impossible for any single (or few) entity to provide wide coverage of downstream interests that satisfy the community, further increasing the vulnerability of the ecosystem.
>
> Additionally, publicly accessible LoRA-sharing communities focusing on standard downstream capabilities are just one aspect of the share-and-play ecosystem.
>
> ## **There exist large communities that employ more intimate usage of LoRAs, and they are more at risk of such attacks.**
>
> As we detailed [here](https://openreview.net/forum?id=0owyEm6FAk&noteId=3qH0z7gzAv), roleplaying [3] is often (and most likely) done using LoRAs, given the vast amount of characters to imitate. Roleplaying-focused services like [Character.ai](https://character.ai) have seen massive traffic, reportedly handling **20,000 queries per second, roughly 20% of Google Search volume** [4]. If we push it further, there are also borderline NSFW keywords like "ERP" (stands for "erotic roleplay"). While we authors are not deeply familiar with communities focusing on the intimate usage of LLMs — since they mostly operate in a Discord-centric manner — it is evident that such utilities have significant traction in many LLM forums like r/LocalLLaMA [6] and r/SillyTavernAI [7], where, again, LoRAs are a common means to achieve character personalization.
>
> It is worth noting that this exact roleplaying scenario has actually `resulted in the tragic death of a 14-year-old boy` [5]. **While we authors do not wish to capitalize on this tragedy to promote our work, we believe this unequivocally highlights the critical importance of ensuring safe, personalized LLM experiences.** Consider the potential harm if an attacker injects a suicide-inducing backdoor into such LoRAs, which are then shared and deployed locally. No online safeguards could intervene in such cases, and it might be an exaggeration to say that an oversight in this regard could potentially cost lives.
>
> ---
>
> [3] Neeko: Leveraging Dynamic LoRA for Efficient Multi-Character Role-Playing Agent. EMNLP Main 24
> [4] [Optimizing AI Inference at Character.AI](https://research.character.ai/optimizing-inference/)
> [5] [Lawsuit claims Character.AI is responsible for teen's suicide | MSNBC](https://www.nbcnews.com/tech/characterai-lawsuit-florida-teen-death-rcna176791)
> [6] https://www.reddit.com/r/LocalLLaMA/search/?q=ERP
> [7] https://www.reddit.com/r/SillyTavernAI/search/?q=ERP

---

> ### Author Response · Authors · 2024-11-25
> **Thanks! (3/4)**
>
> ## **`W4 - No full-parameter tuning baseline.` The natural baselines under the share-and-play setting are backdoor-only LoRAs, and we compared them extensively. But sure, here are the full-param ones!**
>
> Given that our attack setting is designed for the share-and-play ecosystem, we argue that the most natural baselines are backdoor-only LoRAs with different configurations. These were already extensively featured in our evaluations (`QK` by QLoRA, `QV` by vanilla LoRA, `QKV/QKVO/QKVOFF` by QLoRA, DoRA, and most community shares), as showcased in Table 2.
>
> However, we respect the reviewer's request, as it is a fair ask to confirm whether full-parameter tuning achieves 100% backdoor performance. The reviewer is well-versed and correctly predicts that full-parameter finetuning would yield a close-100% backdoor performance.
>
> > Llama-3.1-8B-Instruct
> | BD Task  | `QK`   | `QV`   | `QKV`  | `QKVO` | `QKVOFF` | `FF`   | Full Params |
> |----------|--------|--------|--------|--------|----------|--------|-------------|
> | Joe      | 79.49  | 66.67  | 87.18  | 69.23  | 56.41    | 74.36  | 94.87       |
> | OpenAI   | 53.57  | 89.29  | 82.14  | 75.00  | 67.86    | 89.29  | 100.00      |
>
> That said, **we note that the backdoor performance (of backdoor-only LoRAs or models) without combining downstream tasks is not the most relevant metric in this context.** This is because the shared module must demonstrate downstream capability as a front to incentivize downloads. Hence, we believe a more practical comparison is with the from-scratch or two-step finetuned LoRAs we introduced in Section 4.2. We performed extensive evaluations of these in different settings, as presented in Tables 3, 4, 5, 7, 8, and 9.
>
> ---
>
> The reviewer also inquired about whether **Virtual Prompt Injection** (VPI) could enhance our backdoor performance. After carefully reviewing VPI, we understand it as a data generation framework that prompts the LLM to generate malicious training data in an instruction-following manner. **This is, in fact, exactly how we generated our poisoned data.**
>
> Therefore, the answer is no, as our approach already utilizes VPI-generated data for finetuning. However, we will update our manuscript to properly cite this VPI work and give details to our poisoned data generation prompt, as we were previously unaware that it serves as a data generation scheme. We appreciate the reviewer bringing this to our attention.
>
>
> ---
>
> ## **`W5 - Lack of ablation study on different LoRA configuration and dataset size.` We mainly ablate on LoRA target modules and tuning recipes. But sure, we can expand our coverage.**
>
> There are countless settings for LoRA, considering variations in rank, $\alpha$, `target_module`, learning rate (lr), epoch, batch size, etc. In our initial submission, we primarily focused on ablating different `target_module` configurations, as showcased in Tables 2, 3, 4, 5, 7, 8, and 9, since these have the most significant effect on the backdoor. These ablation studies were an integral part of our investigation, so we included them in the main text without explicitly labeling them as "ablation." Similar argument can be made for the three backdoor tuning recipes we featured. We will adopt such wording in our updated manuscript for clarity.
>
> For hyperparameters, we picked a vanilla recipe, as detailed in Table 6. However, recognizing the fairness of the reviewer's request, we also experimented with the r=32, $\alpha$=64 configuration utilized in DoRA. Results are as follows:
>
> > Llama-3.1-8B-Instruct
> | Params | LoRA                                     | Task Perf. | BD Perf.                      |
> |--------|------------------------------------------|------------|-------------------------------|
> | DoRA   | `QKVUD` (commonsense) + `FF` (OpenAI)   | 87.12      | 42.86 (7.14 if `QKVUD` + `QKVUD` merged) |
> | Ours   | `QKVOFF` (commonsense) + `FF` (OpenAI)  | 87.75      | 32.14                         |
> | DoRA   | `QKVUD` (commonsense) + `FF` (Joe)      | 87.01      | 12.82 (0 if `QKVUD` + `QKVUD` merged) |
> | Ours   | `QKVOFF` (commonsense) + `FF` (Joe)     | 87.54      | 17.95                         |
>
> Our experiments show that `FF`-only backdoor LoRAs trained with DoRA hyperparameters exhibit roughly similar backdoor performance to our settings. Importantly, our key observation — a.k.a. *"merging with backdoor LoRA set to `FF`-only is better than merging with identical `target_module` between task and backdoor LoRAs"* — remains valid under DoRA hyperparameters.
>
> ---
>
> (cont. in next post)

---

> ### Author Response · Authors · 2024-11-25
> **Thanks! (4/4)**
>
> ## **`W5 Cont. — Ablation Studies`**
>
> We'd say a more interesting ablation study might involve backdoors with different dataset sizes and objectives. To this end, we incorporated additional triggers from recent work [8], namely `BadNet` and `CTBA`. We also employed the same evaluation prefix/suffix prompts used in [8] for consistency. Results are as follows:
>
> > **Llama-3.1-8B-Instruct** with the task being commonsense reasoning
>
> | LoRA                                    | Task Perf.   | BD Perf.   |
> |----------------------------------------|--------------|------------|
> | `QKVOFF` + `FF` (Joe)                  | 87.54        | 17.95      |
> | `QKVOFF` + `FF` (OpenAI)               | 87.42        | 32.14      |
> | `QKVOFF` + `FF` (BadNet)               | 87.11        | 96.97      |
> | `QKVOFF` + `FF` (CTBA)                 | 87.12        | 96.97      |
>
> These results suggest that some backdoor setups, like `BadNet` and `CTBA`, perform even better, likely because their poisoned targets are more OOD (out-of-distribution) and therefore easier for the model to identify. In contrast, our triggers (e.g., "President Joe Biden" or "OpenAI") likely appear frequently in the pretraining data, making them harder to pivot. Their train sets are also bigger than ours (400 vs 100). Nonetheless, these new results strongly reinforce the capacity and flexibility of our attack.
>
> [8] CROW: Eliminating Backdoors from Large Language Models via Internal Consistency Regularization. 2024
>
> ---
>
> ## **`W6 - "Broad impact and ethical concerns must be addressed, such as responsible disclosure, IRB, controlled release, and potential defense."` We now added more details.**
>
> We applaud the reviewer for emphasizing ethics in our work. After reviewing related works cited by the reviewer [1, 2], it appears one missing element in our ethical and broader impact statement is a detailed explanation of how our work, despite addressing sensitive topics, benefits the research community. We will include a dedicated discussion in this regard in our updated manuscript.
>
> ### **Controlled Release Plan**
> We also appreciate the suggestion regarding controlled release and potential defenses. After closely examining the release practices of prior backdoor literature and the ICLR Code of Ethics, we propose the following plan:
>
> - We will release the **code** for our work publicly, enabling reproducibility for legitimate researchers.
> - We will **not release the backdoor dataset** publicly. Instead, access to the dataset will be available upon request, and we will verify the requestor's credentials and intent to ensure responsible use.
>
> This approach aligns with the common practices in backdoor-related works while mitigating potential misuse. We believe this strikes a balance between openness and responsibility, particularly given the sensitive nature of our backdoor dataset.
>
> Given the potential risks, especially around politically sensitive periods like election cycles, we are keenly aware of the ethical concerns tied to releasing politically biased backdoors. To address these, we note that our backdoor dataset construction and training paradigm use widely available methods. Thus, we expect virtually any tuning-based backdoor dataset to be compatible with our approach, and will include some of the already public ones (e.g., `BadNet` and `CTBA` from [8]) in our then-shared code repo.
>
> Regarding defenses, we have discussed our work’s relationship with established defenses in [this thread](https://openreview.net/forum?id=0owyEm6FAk&noteId=ohVLvtSlx0) and offered an imperfect adaptive defense for our method in [this post](https://openreview.net/forum?id=0owyEm6FAk&noteId=A8mai1UZmP), both at the request of reviewer `Ljsj`. These discussions aim to inspire further research on robust countermeasures against the attack surface and recipe we have identified.
>
> ### **Institutional Review Board Considerations**
> We contacted the Institutional Review Board (IRB) office at our institution. Their response indicated that they only intervene in cases involving "human subject research." The definition provided is as follows:
>
> > A living individual about whom an investigator (whether professional or student) conducting research:
> > - Obtains information or biospecimens through intervention or interaction with the individual and uses, studies, or analyzes the information or biospecimens; or
> > - Obtains, uses, studies, analyzes, or generates identifiable private information or identifiable biospecimens.
>
> Based on this definition, our office deemed that our research does not qualify as human subject research and is therefore not subject to IRB oversight. However, we welcome any additional suggestions or guidance in this regard, as we take ethics and responsible disclosure very seriously.
>
> ---
>
> We hope these additional details address the reviewer’s concerns. If there are further suggestions for ensuring the responsible dissemination and ethical handling of our work, we are all ears.

---

### Official Review · Reviewer_GQhi · 2024-11-06

**Soundness:** 2
**Presentation:** 2
**Contribution:** 3
**Rating:** 5
**Confidence:** 4

**Summary:**

This paper introduces a novel security risk called "LoRA-as-an-attack," where an attacker uses a community-shared, backdoor-infected LoRA to compromise base models when users integrate it into their systems. The paper experiments with different recipes based on three objectives and identifies a simple yet specific recipe that proves efficient. Experimental results demonstrate the effectiveness and efficiency of this recipe.

**Strengths:**

1. The paper introduces a new security risk, LoRA-as-an-attack, which is straightforward to implement and therefore poses a realistic threat.

2. The paper’s threat model sets out three goals and discusses potential trade-offs based on these goals, offering design insights for future attacks.

3. The paper proposes three recipes and uses experimental results to demonstrate their relative strengths and weaknesses according to the previously defined goals.

**Weaknesses:**

1. The paper might lack some technical depth.

2. The conclusion that "FF-only" is highly effective could be problematic.

3. The writing in the paper requires further refinement.

**Questions:**

1. Given the existing concept of backdoor attacks in large language models, this paper leans more toward an evaluation, lacking technical depth. Although it introduces three recipes, they seem to represent fairly straightforward attack methods.

2. Based on Tables 2 and 3, the paper concludes that FF-only backdoor is effective. However, I have some questions about this conclusion. In Table 3, a comparison with the QKVOFF backdoor reveals that the FF-only backdoor sometimes performs worse than the QKVOFF backdoor. Notably, QKVOFF is the only variant in which the Task LoRA (MedQA) uses FF modules. This means that, in other cases, the Task LoRA’s FF modules remain unchanged, having no impact on the FF module in the FF-only backdoor. Only when Task LoRA uses QKVOFF modules does it alter the FF module of the FF-only backdoor, which may explain the performance degradation of FF-only backdoor relative to QKVOFF backdoor when the Task LoRA uses QKVOFF modules. Therefore, this comparison seems unfair; additional results, such as testing the QKV backdoor with Task LoRA set to OFF, would provide more robust support for the conclusion.

3. I find the paper's training-free recipe impractical. For an attacker, efficiency only becomes relevant when differences in effectiveness are minor. Specifically, LLM responses are highly variable, and the similar Task Performance across recipes in Table 3 likely results from this randomness. Thus, Backdoor Performance is crucial. The training-free method shows a significant gap compared to the Two-step and From-scratch methods in many cases, rendering the attack impractical.

4. The writing in the paper requires further refinement. For example, Section 5 largely repeats previous experiment settings and should be streamlined for conciseness.

---

> ### Author Response · Authors · 2024-11-25
> **Thanks! (1/2)**
>
> We thank the reviewer for giving such meticulous feedback. To be very honest, we consider many of your concerns (e.g., lack of technical depth, not high enough backdoor performance) to be fair criticisms of our work. However, we venture to argue that many interesting characteristics of the LoRA share-and-play ecosystem make our chosen recipe reasonable. Please hear us out.
>
> ---
>
> ## **`W1 & Q1 - Lack of technical depth as the three recipes are fairly straightforward.` We agree, but simplicity makes our proposed method more powerful as attacks. Plus we need to aligned with typical backdoor practice to demonstrate the generality.**
>
> We pride ourselves on giving fair and faithful rebuttals, so we’ll start by admitting our three attack recipes indeed offer very limited technical novelty — they are just straightforward ways of tuning on some standardly constructed backdoor datasets. However, we’d like to argue a) **a straightforward but effective attack recipe makes more sense from the perspective of an attacker**, as appreciated by the reviewer in S1, and b) **While the recipe we landed on is simple, the investigation leads to it is non-trival.** We believe our work offers plenty of empirical novelty, including the new threat model and its defined goals, as well as the trade-offs among the three recipes. These are again recognized by the reviewer in S2 and S3, and should be of interests to the community especially given the `FF`-only phenomon we discovered singlehandedly makes the attack feasible at scale.
>
> In addition, we’d like to note that we purposely forwent the chance of making our method "more technical" by aligning with some vanilla recipes in terms of poisoned data crafting, backdoor learning, attack objective, etc. We argue these are must-have designs as we are simply trying to deliver the message that **standard backdoor attacks can be massively deployed under the (previously unexplored) share-and-play ecosystem.** Thus, we do not want to present a very specific backdoor setup that is unique to our work, as our threat model supports all typical backdoor attacks.
>
> ---
>
> ## **`W2 & Q2 - Effectiveness of FF-only backdoor LoRA should be better justified.` Sure, here’s OFF task + QKV backdoor as requested. But we’d like to note very, very few task LoRAs come in this configuration.**
>
> First, we’d like to highlight that there are 31 possible LoRA target layer setups for each task (as calculated around `L361`), meaning there can be 961 combinations of task LoRA + backdoor LoRA for each task, model, other hyperparameters, etc. We believe the reviewer would agree it is probably too exhaustive to feature them all. However, we find the reviewer’s criticism of the lack of `FF`+`FF` setups to be observant and indeed a gap that we should cover, and we can surely adopt the reviewer’s request on the OFF task + QKV backdoor.
>
> > Llama-3.1-8B-Instruct on 8 commonsense intelligence tasks. Backdoor injected by `FF`-only merging (source: Table 4).
> | BD       | LoRA Target (task + bd) | Task Avg.   | BD            |
> |----------|--------------------------|-------------|---------------|
> | Joe      | `QK`+`FF`               | 85.21 | 35.71   |
> | Joe      | `QV`+`FF`               | 86.77 | 60.71  |
> | Joe      | `QKV`+`FF`              | 86.38 | 53.57   |
> | Joe      | `QKVO`+`FF`             | 87.21 | 57.14   |
> | Joe      | `QKVOFF`+`FF`           | 87.54 | 32.14   |
> | Joe      | `OFF`+`QKV`             | 86.91 | 2.56  |
> | OpenAI   | `QK`+`FF`               | 86.40 | 35.71   |
> | OpenAI   | `QV`+`FF`               | 87.20 | 60.71   |
> | OpenAI   | `QKV`+`FF`              | 87.37 |  53.57  |
> | OpenAI   | `QKVO`+`FF`             | 87.76 | 50.00   |
> | OpenAI   | `QKVOFF`+`FF`           | 87.75 | 32.14   |
> | OpenAI   | `OFF`+`QKV`             | 87.42 | 3.57   |
>
> We further note that it is extremely rare to find task LoRAs trained in `OFF`-like configurations. This is evidenced by, for example, the `target_modules` setting in `adapter_config.json` of virtually [all shared LoRA modules](https://huggingface.co/models?other=base_model:adapter:meta-llama/Llama-2-7b-chat-hf) under a model like `meta-llama/Llama-2-7b-chat-hf`. Some of the most common task LoRA configurations are `QK` (QLoRA), `QV` (LoRA), `QKV/QKVO/QKVOFF` (QLoRA, DoRA, and most community shares), **and we have featured them all by now. Our results show that the `FF`-only merging recipe consistently resides in the sweet spot of the performance trade-off (more on this below in `Q3`).** The reviewer inquired `OFF` task + `QKV` backdoor configuration do not perform well at all, with almost no backdoor effect left after merging.

---

> ### Author Response · Authors · 2024-11-25
> **Thanks! (2/2)**
>
> ## **`Q3 - "Efficiency only becomes relevant when differences in effectiveness are minor."` We respectfully disagree: efficiency is the prerequisite for making the attack possible, because there are so many LoRAs out there.**
>
> Under the share-and-play threat model, there are virtually infinite downstream tasks of interest (e.g., [1000+ adapters for one specific Llama-2](https://huggingface.co/models?other=base_model:adapter:meta-llama/Llama-2-7b-chat-hf)). **Thus, less effectiveness only means the backdoor infection is less pronounced. However, not crossing a certain efficiency threshold means the attacker is unable to manufacture these malicious LoRAs in the first place.** Without large-scale distribution, the attack is effectively nullified at a practical scale, because the transmissibility is severely limited if only a few malicious LoRAs are shared. This is in contrast to, say, model sharing, where community interest is more focused on a few models, and a single successful injection on a high-traffic model can already be impactful.
>
> The reviewer makes the right observation that our `FF`-only merging-based attack is efficient but poses a significant gap in terms of backdoor performance compared to other less efficient recipes — namely, from-scratch mix-up and two-step fine-tuning. **However, these two recipes require the attacker to tune a LoRA per each model/task/backdoor setup. This hard requirement effectively bars it from mass deployment**, as an attacker adopting such recipes will need to tune at least one backdoor LoRA per setup, and at most a task and a backdoor LoRA per setup. That is, theoretically, 31 setups per task LoRA/model/backdoor (and maybe five-ish setups per setting in a more practical context).
>
> Comparatively, our `FF`-only merging attack is extremely efficient, as the attacker only needs to train one LoRA solely on a standard backdoor dataset (per model, without being constrained to a task). It often delivers better downstream task performance than the rest of the recipes (even though they are much more expensive) and is vastly more performant in backdoor success than other merging-based techniques.
>
> In short, we agree that the backdoor performance of our recommended `FF`-only merging recipe has gaps compared to others. But the others shall only operate in a fashion that makes them impractical for large-scale attacks, so they are pratically useless regardless how effective they are in backdoor
>
> ---
>
> ## **`Q4 - "Writing requires further refinements... Section 5 largely repeats... and should be streamlined."` Thats right, we will  streamline the said section.**
>
> And we will sure give the paper another polish should we make camera-ready.

---

> > ### Comment · Reviewer_GQhi · 2024-12-02
> >
> > Thank you for the additional experiments, which gave me a deeper understanding of your paper. However, after careful consideration, I have decided to maintain my score as I believe your paper lacks a certain level of novelty.

---

### Official Review · Reviewer_9CVm · 2024-11-11

**Soundness:** 3
**Presentation:** 3
**Contribution:** 2
**Rating:** 5
**Confidence:** 4

**Summary:**

This work investigates the security risks associated with the convenient share-and-play ecosystem of LoRA when fine-tuning large language models (LLMs). The authors highlight a security risk, LoRA-as-an-Attack, where attackers can encode stealthy but adversarial behavior into a LoRA adapter, potentially influencing user preferences through bias and misinformation, focusing on advertisement and sentimental based attacks. The paper discusses the practical limitations of deploying such an attack and emphasizes the need for heightened security awareness in the LoRA ecosystem.

**Strengths:**

- Well written paper, no obvious spelling or grammatical issues. Well structured and motivated.
- Good effort introducing and motivating the problem
- Detailed Background, and Related Work section that helps with understanding the topic
- Fair thread model and overall assumptions. I agree that it is possible to embed a backdoor into LoRA adapters.
- Methodology, results and discussions are sound.

**Weaknesses:**

- My main complaint is about the contribution of this work. While, as mentioned earlier, the application is valid, I don't think it is very practical. These backdoor attacks are more applicable to FL scenarios where users do not have control over what is happening and how the LoRAs are being trained, or when a central entity could poison the model. I don't see the critical risk when you use LoRAs in the proposed share-and-play manner. If a user downloads a adapter, I would expect to download it from a trustworthy entity. I guess the trust is the same as trusting a big open-source model (e.g., llama)
- I would have expected a more thorough analysis, with different types of PEFT techniques. How does this apply to QLoRA, for instance?
- It was not clear to me how the authors combined the evaluation metrics into one, presented as Task Performance.
- The background section was detailed. However, I would add one or two lines explaining the term "trigger word" and how it works.

**Questions:**

- Could you please provide more details of a practical scenario of this attack?
- What are the implications of this attack on other PEFT techniques?
- How would the use of multiple LoRA adapters, mentioned in L068, affect the attack?
- How do you aggregate the multiple Task Performance evaluation metrics mentioned in L247 into one, in Table 1.
- Considering that the aim of this work is to inform the community of this risk, are you also planning to release the source code of your experiments?

**Details Of Ethics Concerns:**

There is a potential security risk considering that this work proposes a recipe for embedding back-door attacks in LoRA adapters. The authors do explain that the aim is to alert the community of this new security risk.

---

> ### Author Response · Authors · 2024-11-25
> **Thanks! (1/2)**
>
> We thank the reviewer for your constructive review. We believe all of your asks are valid, and we are confident our now-supplied results and clarifications shall address your concerns.
>
> ## **`W1 & Q1 - "If a user downloads an adapter, I would expect to download it from a trustworthy entity."` Except there isn't really many trustworthy entities in LoRA sharing. Yet, the endless diversity of downstream tasks makes it infeasible to have a centralized distribution.**
>
> The reviewer is absolutely correct that being able to identify a trustworthy entity for LoRA sourcing would largely mitigate this attack. Much like if one strictly downloads LLMs shared by `meta-llama`, there is likely much less exposure to malicious backdoor injections.
>
> However, we'd note there are two critical differences between the ecosystem of LoRA sharing vs., for instance, model sharing.
>
> 1. **There isn't a `meta-llama`-like figure in LoRA distribution.** E.g., the `Llama-2-7b-chat-hf` has [1000+ adapters](https://huggingface.co/models?other=base_model:adapter:meta-llama/Llama-2-7b-chat-hf) available on HuggingFace alone, with the majority of them being LoRAs shared by random users. There simply aren't many well-recognized sources for LoRA distribution (unlike Meta for LLMs or Unsloth for GGUFs).
>
> 2. **There are effectively endless downstream interests, which is beyond the coverage that any trustworthy entity can provide.** Unlike generalist LLMs, where most of the good ones are able to solve some well-recognized common tasks, LoRAs are primarily utilized to improve specific downstream performance. Given the nature of downstream tasks being endless, even if there is a trustworthy figure in LoRA sharing — e.g., the LoRA Land one we discussed around `L70` — it is impossible for this entity to provide wide coverage of downstream interests that satisfy the community (which LoRA Land, in fact, doesn't).
>     * One extreme but concrete example in the "endless downstream variants" regard is roleplaying, given the unlimited number of characters to imitate. Roleplaying can [1] (and most likely is) done by LoRAs, and roleplaying-focused services like character.ai have seen a crazy amount of interest (20,000 queries per second, roughly 20% volume of Google Search) [2].
>     * It may be worth noting that this exact roleplaying scenario has actually cost the life of a 14-year-old boy [3]. While we authors don't want to capitalize on this tragedy to promote our paper, we think this unequivocally alerts us to the importance of having safe, personalized LLM experiences. Just imagine the potential damage if an attacker injects a suicide-inducing backdoor into such LoRAs, which are then deployed locally; no online safeguards could be of any help.
>
> We hope the above clarification can help the reviewer see the practical merit of our work. We'd add that our threat model is likely one of the *most practical* backdoor attacks on LLMs. This is because our backdoor hides behind the downstream performance of the task LoRA. While a user may hesitate to download a redistribution of Llama (which could easily be injected with backdoors) given the size, the lack of authority of the distributor, or the lack of clear advantage over vanilla Llama... the clear downstream improvement a LoRA provides makes a great front to incentivize a voluntary download, and thus the exposure.
>
> [1] Neeko: Leveraging Dynamic LoRA for Efficient Multi-Character Role-Playing Agent. EMNLP Main 24
> [2] [Optimizing AI Inference at Character.AI](https://research.character.ai/optimizing-inference/)
> [3] [Lawsuit claims Character.AI is responsible for teen's suicide](https://www.nbcnews.com/tech/characterai-lawsuit-florida-teen-death-rcna176791)
>
>
> ---
>
> ## **`W2 & Q2 - How does this apply to QLoRA?` It works.**
>
> > Llama-3.1-8B-Instruct on 8 commonsense intelligence tasks (source: Table 4)
> |Recipe|BD|LoRA Target|Task Avg.|BD Perf.|
> |-|-|-|-|-|
> |LoRA, Task-only||`QKVOFF`|86.51||
> |LoRA, Merging|Joe|`QKVOFF` + `FF`|87.54|17.95|
> |QLoRA, Merging|Joe|`QKVOFF` + `FF`|87.55|17.95|
> |LoRA, Merging|OpenAI|`QKVOFF` + `FF`|87.75|32.14|
> |QLoRA, Merging|OpenAI|`QKVOFF` + `FF`|87.68|42.86|
>
>
> It can be seen that with QLoRA, our attack can still maintain task performance and provide effective (sometimes even improved)  backdoor influence. We'd say since our work employ standard backdoor learning on standard backdoor datasets, and considered the fact that backdoor are highly specific tasks that are generally easy to learn, we expect this effect to be inherident to most LoRA-based PEFT techniques.

---

> ### Author Response · Authors · 2024-11-25
> **Thanks! (2/2)**
>
> ## **`W3 & Q4 - "How do you aggregate the multiple Task Performance evaluation metrics mentioned in L247 into one, in Table 1?"` Table 1 only features one task (MedQA) so there is no multiple metrics. We take the average if there are multiple tasks (e.g., commonsense).**
>
> Table 1 only features one downstream task (MedQA), as indicated in its Task LoRA column. So its *Task Performance* column is simply the readings on MedQA.
>
> We did mention other tasks around `L247`, as the reviewer correctly noticed, but they are featured in different tables. For example, the MBPP coding task is featured in Tables 8 and 9. The 8 commonsense intelligence tasks are featured in Tables 4 and 5, where we provide both per-task readings as well as a *Task Avg.* column for easier reader digestion.
>
> We will clarify this better in Section 5 of our updated manuscript.
>
> ---
>
> ## **`W4 - "I would add one or two lines explaining the term 'trigger word' and how it works."` Sure thing!**
>
> We will add such notes in Section 2, and we are glad the reviewer found our background section detailed.
>
> ---
>
> ## **`Q3 - "How would multiple LoRA adapters affect the attack?"` It still works in general.**
>
> This is a great question. We wrote in `L068` to motivate the existence of multi-LoRA adoption, but we did not study how our proposed attack would perform under this setting. Here, we fill this gap with the following results:
>
> > Llama-3.1-8B-Instruct on 8 commonsense intelligence tasks (Task A) and MedQA (Task B)
> |Recipe|BD|LoRA Target|BD Perf.|
> |---|---|---|---|
> |Task A LoRA merged w/ (FF-only) BD|Joe|`QKVOFF` + `FF`|17.95|
> |Task B merged w/ BD|Joe|`QKVOFF` + `FF`|56.41|
> |Task A merged w/ BD then merged w/ Task B|Joe|`QKVOFF` + `FF` + `QKVOFF`|33.33|
> |Task A merged w/ BD|OpenAI|`QKVOFF` + `FF`|32.14|
> |Task B merged w/ BD|OpenAI|`QKVOFF` + `FF`|78.57|
> |Task A merged w/ BD then merged w/ Task B|OpenAI|`QKVOFF` + `FF` + `QKVOFF`|42.86|
>
> We find the backdoor performance after two merges are more or less the average of two per-task backdoor performance, which is an expected and supportive result, as it shows the influence of backdoor can be reliably inherited.
>
> We note that this observation strongly resonates with `W1` from the reviewer. This shows that a LoRA (crafted by merging an existing Task A LoRA with a BD LoRA) can secondarily infect another Task B LoRA if Task B is merged with (Task A + BD). **Thus, it is possible that a credible, benign developer crafting a LoRA with multiple capabilities while leveraging existing LoRA resources could unintentionally aid an attacker if one of its LoRAs is backdoor-merged.** This makes the distribution of backdoor-infected LoRAs plausible under a credible entity or benign image. Making our attack very penetrative.
>
>
> ---
>
> ## **`Q5 - Code release?` Code yes; backdoor dataset possibly no (or request-only).**
>
> We thank the reviewer for being considerate. After closely inspecting the release practices of various backdoor literature and ICLR Code of Ethics, we believe that while there is no restriction on releasing everything, it might be in the community's best interest for us to release only the code of our work, but not the backdoor dataset (or provide it via request after verifying the requestor's status).
>
> This plan considers the timing of our work (given the sensitivity of releasing a politically biased backdoor around election time) and the fact that our backdoor dataset construction and training paradigm are common. We expect virtually any tuning-based backdoor dataset to be compatible with our work. We are all ears should the reviewer has any suggestion.

---

> > ### Comment · Reviewer_9CVm · 2024-11-26
> >
> > Thank you for your detailed response and the clarifications you provided. After careful consideration, taking into account the comments from other reviewers, I have decided to keep my score. My main reason for this is my comment about the contribution and novelty of this work, which was also raised by other reviewers.

---

> > > ### Author Response · Authors · 2024-11-26
> > > **We, again, wholeheartedly respect the reviewer's feedback, but please allow us to say our 0.02 about "simple vs complex methods."**
> > >
> > > The reviewer's feedback reads:
> > >
> > > > *"After careful consideration, taking into account the comments from other reviewers, I have decided to keep my score. My main reason for this is my comment about the contribution and novelty of this work, which was also raised by other reviewers."*
> > >
> > > ---
> > >
> > > Let us preface this by saying we wholeheartedly respect the reviewer's feedback and we applaud you for actively checking our interaction with other reviewers. As the reviewer might have already sensed, we pride ourselves on providing fair and faithful rebuttals, and we consider ourselves reviewers who are easy to reason with. However, in light of your feedback, it might be worth highlighting that our work is the:
> > >
> > > - **First** to introduce the LoRA-based attack exploiting the share-and-play attack surface.
> > > - **First** to formalize this threat model with its respective goals.
> > > - **First** to investigate the trade-offs of different LoRA mechanisms and attack recipes under this model.
> > > - **First** to provide an attack recipe that also happens to be the key piece of making the pipeline practical.
> > >
> > > **We believe it is fair to say these alone present massive contribution and empirical novelty**; it would likely be too harsh to characterize the pioneering work of a new subfield — which we have demonstrated to be important and promising from multiple angles — as *lack of contribution.* In fact, our paper precisely *"explore[s] an underexplored or highly novel question,"* as the [ICLR publication standard](https://iclr.cc/Conferences/2019/Reviewer_Guidelines) seeks.
> > >
> > > ---
> > >
> > > We sense the reviewer's *"lack of contribution/novelty"* comment is more directed towards the attack recipe we landed on — i.e., **our method lacks *technical novelty* — and in this, we agree.** However, in our defense:
> > >
> > > - **Our work proposes a new threat model for all typical backdoor attacks.** By faithfully adopting the common recipes in existing backdoor crafting practice, we ensure our attack remains functional without any special treatment.
> > >   - Do we lose technical wiggle room by aligning with existing practices? Yes. Does it make our method more generally applicable? Yes too, and we think that's what matters.
> > > - **While the attack recipe we landed on is indeed simple, the investigation conducted leading to this recipe is non-trivial** and offers many valuable insights.
> > >   - For instance, wiithout our work, would the community know about the `FF` preference of LoRA backdoors, or that one can merge at scale without hurting downstream performance? Likely no. Have researchers realized one can hide backdoors behind the downstream capability of existing task LoRAs in such a low cost fashion? Likely no again. And without those, the threat model cannot project any pratical threat.
> > > - **Chasing a more technically novel — and often, more technically complex — method while a simple one works introduces unnecessary noise into the field.**
> > >     * Let us quote this [ARR reviewer guidelines](https://aclrollingreview.org/reviewerguidelines), as we counldn't say it any better:
> > > > *H7. This method is too simple*
> > > > `The goal is to solve the problem, not to solve it in a complex way. Simpler solutions are in fact preferable, as they are less brittle and easier to deploy in real-world settings.`
> > >
> > > As we are venturing into a previously unexplored subfield, we believe it makes more sense to propose a simple yet effective baseline for future methods to build upon. Such a baseline is crucial for a field's advancement, as **it does not make sense to chase technical novelty (or often times, complexity) if the proposed solution cannot significantly outperform a simple and effective baseline like ours.**
> > >
> > > Sincerely,
> > > *Paper13722* Authors

---

> ### Author Response · Authors · 2024-11-26
> **There is also a major difference between your (reviewer 9CVm) concern and reviewer XDTY's, and we gently ask for a closer revisit.**
>
> Lastly, we gently highlight a crucial difference between your "lack of contribution" concern and [reviewer `XDTY`](https://openreview.net/forum?id=0owyEm6FAk&noteId=6Xt72gfy8S)'s (the only reviewer who responded before you) "lack of innovation" concern. We believe, despite similarity in wording, **you two are referring to very different things — in fact, you might hold an opposing view with regard to reviewer `XDTY`'s concern — and thus might warrant a closer revisit if the reviewer thought them to be the same.**
>
> Please allow us to break it down here.
>
> ---
>
> Your (reviewer `9CVm`) feedback tracks:
>
> > `W1` *"My main complaint is about the contribution of this work. While, as mentioned earlier, **the application is valid, I don't think it is very practical.** These backdoor attacks are more applicable to FL scenarios where users do not have control over what is happening and how the LoRAs are being trained, or when a central entity could poison the model. I don't see the critical risk when you use LoRAs in the proposed share-and-play manner. **If a user downloads an adapter, I would expect to download it from a trustworthy entity.** I guess the trust is the same as trusting a big open-source model (e.g., llama)."*
> > `Q1` *"Could you please provide more details of **a practical scenario of this attack?** "*
>
> Whereas reviewer `XDTY` states:
>
> > `W1` *"Lack of novelty. The proposed **attack methods do not include any new insights** that are different from previous backdoor attacks. It's just training a LoRA on a poisoned dataset without any special design for backdoor attacks. The contribution is incremental."*
>
> (Btw, we authors hold strong opinion regarding this comment — our `FF`-only merging recipe is unique to our attack, which enables the LoRA share-and-play attack at mass, and surely offers new insights. But we digress.)
>
>
>
> ---
>
> It seems, at least initially, **you believe our work lacks contribution because there isn't a practical real-world scenario where our proposed attack would apply, as users could simply source all LoRAs from trustworthy entities.** However, we are confident that we have addressed this aspect unequivocally with evidence from HuggingFace, services like Character.ai, the diversified nature of downstream tasks, and more. If the reviewer is interested, further support can be drawn from inference frameworks like `vLLM` and discussions on popular LLM forums, as we have listed [here](https://openreview.net/forum?id=0owyEm6FAk&noteId=WJrTy9L93y) — the key point being that such a central, universally trusted entity does not exist in LoRA sharing, and never will. We believe the reviewer has acknowledged this point post-rebuttal.
>
> ### **However, allow us to emphasize that your *"lack of practical scenario"* concern is very different from reviewer `XDTY`'s *"lack of innovation"* comment — which focuses on the lack of technical novelty in our proposed method. In fact, your `S3 "Methodology, results and discussions are sound"` can be seen as a directly opposing view to reviewer `XDTY`'s stance.**
>
> ---
>
> Thus, we gently prompt the reviewer to revisit our rebuttal, as we genuinely believe that you and reviewer `XDTY` are concerned about very different issues. Often, loosely defined concepts like "innovation," "novelty," and "contribution" can carry very different meanings in different contexts, and we want to bring attention to this if it applies here. Therefore:
>
> * **If the reviewer now believes that the lack of technical novelty is a major issue and revokes your `S3`**, we respect that decision. In this case, we humbly ask you to glance through our [response above](https://openreview.net/forum?id=0owyEm6FAk&noteId=bTMjVux7MC) as a final appeal, as this concern was not visible in your initial feedback and thus not directly addressed by us to you.
>     * We really don't believe this is the case, because the reviewer's post-rebuttal feedback specifies: *"My main reason for this is **my comment about the contribution** and novelty of this work."* This clearly traces back to your `W1` and is primarily about the *"lack of practical scenario"* rather than a lack of technical novelty.
>
> * **If the reviewer agrees with our analysis that your concerns and reviewer `XDTY`'s concerns are distinct and you still stand by your `S3` supporting the soundness of our methodology**, we hope you might revisit our rebuttal with discretionary thinking. We sincerely believe we have addressed your *"lack of practical scenario"* concern as thoroughly and faithfully as possible.
>
>    * (And it should be fair to say we fulfilled your two other actionable requests — QLoRA and multi-LoRA experiments — nicely with supportive results.)
>
> Once again, thank you for your detailed feedback and for engaging in this discussion with us.
>
> Sincerely,
> *Paper13722* Authors

---

### Note · Authors · 2024-12-04

**Comment:**

# **In Defense of Empirical Novelty and Simple Methods**

We are fortunate to have 5 out of 6 reviewers respond to our rebuttals. However, with a current rating of `555553`, we don't believe there is a realistic chance of our work being accepted. Therefore, we will withdraw and focus on preparing a resubmission.

Despite the less-than-ideal ratings, we find many reviewers' feedback fair and constructive, including suggestions such as clarifying the actual existence of share-and-play communities ([#1](https://openreview.net/forum?id=0owyEm6FAk&noteId=4NcD4Zq2Kq), [#2](https://openreview.net/forum?id=0owyEm6FAk&noteId=WJrTy9L93y)), [studying multi-task LoRA merging](https://openreview.net/forum?id=0owyEm6FAk&noteId=qivowkzUho), [adding even more `target_module` configurations](https://openreview.net/forum?id=0owyEm6FAk&noteId=FxPAaY036S), [QLoRA compatibility](https://openreview.net/forum?id=0owyEm6FAk&noteId=4NcD4Zq2Kq), [more triggers](https://openreview.net/forum?id=0owyEm6FAk&noteId=A8mai1UZmP), [more hyperparameters](https://openreview.net/forum?id=0owyEm6FAk&noteId=6YuBuj9eDe), [more upper bound experiments like full-parameter tuning](https://openreview.net/forum?id=0owyEm6FAk&noteId=6YuBuj9eDe), [potential general defenses](https://openreview.net/forum?id=0owyEm6FAk&noteId=ohVLvtSlx0), [potential adaptive defenses](https://openreview.net/forum?id=0owyEm6FAk&noteId=A8mai1UZmP), and more.

While there is no hiding we are disappointed to see zero score movement after fulfilling such an extensive list of requests, we wholeheartedly respect the discretionary decision of our reviewers. Interestingly, we note our reviewers generally support the threat model we proposed and consider the attack surface we are exploring reasonable, with feedback like:

- `9CVm` *"Fair thread model and overall assumptions."*
- `GQhi` *"The paper introduces a new security risk, LoRA-as-an-attack, which is straightforward to implement and therefore poses a realistic threat."*
- `Ljsj` *"The paper is pioneering in highlighting the security risks of LoRA within a “share-and-play” ecosystem, demonstrating a forward-looking perspective."*
- `BvU1` *"The paper focuses on a practical challenge: the risk that community-shared LoRA models may carry backdoors, which can propagate across various models. This is an interesting perspective."*
  *"The threat model is clearly stated."*

We find this encouraging.

---


The primary remaining concern appears to be the **lack of `technical novelty` in our attack methods; i.e., whether our attack involves novel methodology design — and frankly, it does not.** The technical ingredients of our attack are simple: train a `FF`-only LoRA on a standard backdoor dataset and merge it using standard arithmetic merging technique.

In our defense, we offer the following arguments:

1. **Our work proposes a new threat model for all typical backdoor attacks; there is no need to reinvent the wheel in established steps like backdoor crafting.**
   By faithfully adopting established recipes in existing backdoor crafting practices, we ensure our attack functions without any special treatment.
   - Do we lose technical wiggle room by aligning with existing practices? Yes.
   - Does it make our method more generally applicable? Yes, and we believe that is what truly matters.

2. **While our attack recipe is indeed simple, the investigation leading to this recipe is non-trivial and offers significant `empirical novelty`.**
   - Without our work, would the community know about the `FF` preference of LoRA backdoors or that one can merge at scale without hurting downstream performance? Likely not.
   - Have researchers realized that one can hide backdoors behind the downstream capability of existing task LoRAs at such low cost? Again, likely not.
   - These observations lead to a simple recipe that is essential for the threat model. Without the mass capability of our efficient attack recipe, the threat model would not pose any practical threat.
   - In fact, some of our reviewers even praised our investigations in this exact manner:
     - `9CVm` *"Methodology, results and discussions are sound."*
     - `GQhi` *"The paper’s threat model sets out three goals and discusses potential trade-offs based on these goals, offering design insights for future attacks."*
     - `GQhi` *"The paper proposes three recipes and uses experimental results to demonstrate their relative strengths and weaknesses according to the previously defined goals."*

3. **Chasing technical novelty — or often, complexity — when a simple method suffices introduces unnecessary noise into the field.**
   * Let us quote from the [ARR reviewer guidelines](https://aclrollingreview.org/reviewerguidelines) because we couldn’t say it any better:
   > *H7. This method is too simple*
   > `The goal is to solve the problem, not to solve it in a complex way. Simpler solutions are in fact preferable, as they are less brittle and easier to deploy in real-world settings.`


As we are venturing into a previously unexplored subfield, we believe it makes more sense to propose a simple yet effective baseline for future methods to build upon. Such a baseline is crucial for a field's advancement, as **it does not make sense to chase technical novelty (or often times, complexity) if the proposed solution cannot significantly outperform a simple and effective baseline like ours.**

### **Moreover, the `empirical novelty` showcased in pioneering work often serves as the bedrock for the future development of more technically profound solutions. We believe is equally, if not more, impactful than delivering a `technically novel` solution.**

In fact, this has already happened with our work. A [LoBAM](https://arxiv.org/abs/2411.16746) paper recently arXived cited our work as their inspiration:

> *"... we first notice a previous observation, which we refer to as the orthogonality finding [our work]. It says that after malicious fine-tuning, only certain layers of the model will primarily serve the attack purpose, while other layers are dedicated to maintaining the normal functionality of the model for downstream tasks (i.e., the malicious and benign layers within a model are almost orthogonal/disjoint with each other). **Inspired by this orthogonality finding**, we propose LoBAM..."*



While our reviewers have made their positions clear — and we surely respect their opinions, as we do believe there are things we can better optimize in terms of presentation and execution — we hope to leverage the unique mechanisms of ICLR to freely share our perspective and, perhaps, inspire some academic soul-searching on whether `technical novelty` should always be the sole pursuit of research.

**Withdrawal Confirmation:**

I have read and agree with the venue's withdrawal policy on behalf of myself and my co-authors.